# Features of Soil Organic Carbon Transformations in the Southern Area of the East European Plain

Fedor N. Lisetskii [1], Zhanna A. Buryak [2], Olga A. Marinina [3], Pavel A. Ukrainskiy [2,3,*] and Pavel V. Goleusov [1]

1 Department of Environmental Management and Land Cadastre, Belgorod State National Research University, 85 Pobedy Str., Belgorod 308015, Russia; fnliset@mail.ru (F.N.L.); goleusov@bsu.edu.ru (P.V.G.)
2 Federal and Regional Centre for Aerospace and Ground Monitoring of Objects and Natural Resources, Belgorod State National Research University, 85 Pobedy Str., Belgorod 308015, Russia; buryak@bsu.edu.ru
3 Centre for Validation and Verification of Carbon Units, Belgorod State National Research University, 85 Pobedy Str., Belgorod 308015, Russia; marinina@bsu.edu.ru
* Correspondence: ukrpa@mail.ru

**Abstract:** The active development of the problems related to the assessment of the role of the pedosphere in global climate change involves the possibility of application of the quantitative determination of soil organic carbon (SOC) as one of the indicators of a climatic response. Here, the authors have summarized the results of their own research over many years (1985–2023), comprising more than 500 determinations of SOC within the area of the Chernozem zone, in the south of the East European Plain (Moldova and Bessarabia, southern Ukraine, southwestern Russia), in the context of regional climate differentiation using evaluations of climatic energy consumption for soil formation. The data were structured for each of the regions through the creation of series of agrogenic soil transformations (virgin land, modern-day ploughed land (<100 years), continually ploughed land (>100 years), fallow land of the modern era (n·10 years), and post-antique long-term fallow land). It has been established, by means of statistical treatment of the data, that the intraregional differentiation of the bioclimatic conditions is the key factor determining the SOC content in the top horizon of soils in the south of the East European Plain. The comparison of the SOC content within the five variants of land use demonstrated that all the regions under study differed, with statistical significance only found in a single type of 'continually ploughed land' (>100 years). This fact reflects the leading role of the duration of agrarian loads in agropedogenesis. If the steppe Chernozems even 145 years ago had a SOC content of up to 4%, then the Chernozems in the forest-steppe zone, which used to have habitats with a SOC content of 4–7%, occupied the largest areas, and have now lost 30–40% of the original values in the 0–50 cm layer. Besides the rates of the SOC degradation, which are known and are comprehensively evaluated in the present work, the phenomenon of prograditation was established in certain situations, in particular arising during the rotation regime of land use (from ploughing to fallow fields, and vice versa), which stimulated effective mechanisms of reproduction of organic substances. Thus, in one of the ancient agricultural regions, where in antiquity the land was cultivated by such ancient Greek states as Tauric Chersonesos and the European Bosporos in Crimea, post-antique long-term fallow lands possess higher SOC contents than their virgin analogues. It is not justified to consider virgin lands as absolute references for the evaluation of the humus conditions since the analysis of agrogenic series of Chernozems has corroborated an essential role of the soil organo-mineral matrix in the formation of the carbon protection capacity.

**Keywords:** soil organic carbon; climatic energy consumption; carbon sequestration; carbon pools; carbon stabilization mechanisms; ploughed land; fallow land





## 1. Introduction

Ecosystems drain the atmospheric carbon dioxide, influencing the global carbonic cycle. The carbon is transformed into the biomass, part of which goes into the pool of the

soil carbon, and therefore can be isolated for millennia. On the global scale, soil represents the largest reservoir of organic matter (SOM) and active surface carbon in the biosphere, being composed of organic and inorganic carbon. Soil is the largest source of carbon in the surface ecosystems, storing about 1500 Pg ($1500 \times 10^{12}$ kg) of carbon in its top one-meter layer and retaining over 70% of the surface organic carbon (Corg) stock; thus, it exceeds the content of carbon in plants by approximately three times as much and the stock in the atmospheric basin of carbon by twice as much [1–3].

The pool of SOC is the most active part of the carbon pool of the surface ecosystem, and its small alterations are capable of exerting a serious influence on the global balance of carbon [2,4]. Generally, the exhaustion of the SOC pool can produce an increase in the concentration of $CO_2$ in the atmosphere and deterioration of the soil quality, which are the key moving forces of global warming and land degradation [5,6]. The finely dispersed mineral fraction was identified as an important element of the system of indicators for effective evaluation of SOC in order to characterize the stabilization of organic carbon in most of the soils [7]. The quantitative assessment of SOC can be made more exact by involving climate indicators, as well as information on the land use, soil management, and characteristics of the vegetation. The formation of stable organic carbon depends on the land use and the management methods of agroecosystems [8]. This comprises the basis of the strategies of biological sequestration in the developed agrarian regions of the agricultural zone. The sequestration of SOC in the ploughed soils is a highly promising method for the alleviation of the consequences of global warming [4]. Thus, a better understanding of the SOC dynamics and factors influencing global warming is necessary for management of the SOC in order to alleviate the modern crises of the changing climate and shortages of provisions.

The parties of the 21st Conference (COP21) of the United Nations Framework Convention on Climate Change (UNFCCC) adopted the Paris Agreement and Initiative for Climate Action Transparency, 4p1000 [9], in order to increase the stock of SOC through the introduction of advanced methods for the management of the agricultural economy. With the adoption of these goals, the intention is to reach an annual increase in SOC at the soil depth of 0–40 cm to help ameliorate the consequences of climate change [10,11]. The results of pilot research have allowed outlining the spheres of the management of land use appropriate for Initiative 4p1000. These goals could be reached through the adoption of conservative methods, such as the no-tillage of soil method [12].

Researchers' attention is focused on such changes in the patterns of land use, e.g., expansion of the fallow regime, that are favorable for the accumulation of SOC. Furthermore, as a review of the literature has shown [13–15], it is ascertained that the duration and rate of carbon accumulation in the soil strongly differ depending on the productivity of the regenerative vegetation, physical, and biological conditions in the soil, as well as the previous history of the intake of the organic carbon by the soil and the mechanical impacts (agroturbations).

At the modern level of the advancements of science and technologies in the context of the international goals for the regulation of the peak carbon dioxide emissions and carbon neutrality, new ideas and methods are needed to gain a comprehensive understanding of the processes and mechanisms of the carbon binding in soils. These differ in the genesis and microbiological activity, the increased effectiveness of the soil carbon binding, and the solutions of the ecological problems caused by climate change and human activities [16].

As shown through an analysis of a series of review papers [7,8,17–22], despite the great activity of scientists, there remains a broad spectrum of questions concerned with the practical soil science, the application of data to the economy of the environment, and the sphere of the monetization of ecosystem services (market of carbon units). In addition, there is a series of methodological tasks that it is necessary to solve at the stage preceding the formation of a sound network of carbon testing sites in a country. The number of these tasks can include the necessity of a more active transformation of the chemical concept of the composition and structure of SOM into a biophysical–chemical conception accentuating

the structural–functional characteristics of the dynamics of carbon at the microscale level, as well as evaluation of the volume of organic carbon of different qualitative composition in terms of the ecosystem services. Replenishment of data on the stocks of carbon (volume concentration), data on the SOM quality (rate of stabilization), etc., are also needed.

There are some unsolved key problems in studies of the long-term deposition of carbon in soils and its rapid involvement in intrasoil processes supporting ecosystem functions and soil services. These problems must be included in the research field of new developments. The planned tasks could comprise:

- Mathematical simulation of humus formation processes and loss in the context of the dynamics of the hydrothermal regime;
- A study of the dependencies between carbon flows in the atmosphere–soil body system and the initial carbon content;
- Mathematical simulation of the process of the humic content change in ploughed soils with consideration of the soil/landscape parameters;
- In addition to the focused studies of the formation of humic substances ensuring stabilization of organic matter in the soil, evaluation of other mechanisms providing SOM with integrity, stability, and protection against decay;
- A quantitative assessment of the contribution of the granulometric potential and colloid-mineral fraction in soil mineralogy to humus accumulation;
- Assessment of the role of different levels of the structural soil organization (macroaggregate and microaggregate) in the rate of carbon fixation.

The aim of this work was (i) to summarize, using statistical analysis, the original author's data on the SOC content in the upper soil horizon for five land use options (virgin land, arable and fallow lands of different times) within the Chernozem belt of the south of the East European Plain (Moldova, south of Ukraine, southwest of Russia); (ii) to establish features and evaluate the rate of long-term changes in the Corg content in Chernozems; and (iii) to compare the agrogenic series of Chernozems with different times of arable and fallow regimes based on the study of one of the first steppe reserves.

## 2. Materials and Methods

### 2.1. GIS Mapping of Energy Costs for Soil Formation

According to the bioenergy approach of V. Volobuev [23], the climatic energy consumption for soil formation ($Q$) is determined by the annual radiation balance of the active surface and the amount of atmospheric precipitation. After the correction of the initial equation [23] for the units of the International System of Units (*SI*), the $Q$ value (MJ m$^{-2}$ yr$^{-1}$) can be calculated as follows:

$$Q = 41.868 \left[ \text{RB} \cdot \exp\left( -18.8 \cdot \frac{\text{RB}^{0.73}}{P} \right) \right], \tag{1}$$

where RB is the radiation budge (kcal cm$^{-2}$ yr$^{-1}$) and $P$ is the annual precipitation, mm.

Due to the correction of Formula (1), which takes into account the need to use the RB values in *SI*, it is proposed to calculate the climatic energy costs for pedogenesis using a transformed equation:

$$Q = \text{RB} \cdot \exp\left( -1.23 \cdot \frac{\text{RB}^{0.73}}{P} \right). \tag{2}$$

With a lack of direct observations of RB, its rather close relationship with air temperature values is used. This dependence, obtained for the southern segment of the East European Plain from long-term data from 73 meteorological stations [24], was also used to supplement direct observations of RB:

$$\text{RB} = 122.727 \cdot T + 923.54, \tag{3}$$

where *T* is the average air temperature for the year, °C.

The mapping of bioclimatic conditions was based on hydrothermal data (average annual temperature, total annual precipitation) from 46 meteorological stations. The time series of observations included the period from 1980 to 2021. The areas on the *Q* map were synthesized using the ArcGIS Spatial Analyst software modules. The map-scheme is built by the spline method with tension. Average values of hydrothermal parameters for key areas are calculated using spatial statistics of raster models.

### 2.2. Physical and Chemical Analyzes of Soils

Chemical analyses of soils were performed in an accredited laboratory using methods accepted for soil science. The Corg content by Tyurin's method (by oxidation of the organic substances with a solution of $K_2Cr_2O_7$ in sulfuric acid). Tyurin's method modified by Ponomareva and Plotnikova (the extraction of humic substances was performed using 0.1 and 0.02 normal solutions of the sodium hydroxide) was used to determine the ratio of humic acids: humic acid (HA) and fulvic acid (FA). Mobile humic acids were determined directly in a decinormal alkaline extract. Mortmass carbon content in soils was determined according to State Standard (GOST) 23740–2016 "Soils. Methods of laboratory determination of organic composition". Kjeldahl's procedure was applied to determine the total nitrogen (N). The method of cation exchange capacity (CEC) determination in natural and disturbed soils is based on the national standard (GOST 17.4.4.01–84), which involves the use of the resulting solution with a pH value of 6.5.

The granulometric composition of the soil was determined by the Kachinsky pipette method, the structural–aggregate analysis was performed by the Savinov method, which made it possible to calculate the structural coefficient as the ratio of the mass of aggregates from 1 to 7 mm to the sum of aggregates < 1 mm and >7 mm (Cstr) and the weighted average diameter of water-stable aggregates (d, mm). The assessment of soil microaggregation was carried out using a microscope at a 98-fold magnification in reflected light, which made it possible to determine the proportion of microaggregates and minerals (elementary soil particles—ESP). The data were included in the microaggregation coefficient (*AC*, %) calculation:

$$AC = 100 \cdot (S - R)/S, \tag{4}$$

where *S* is the proportion of aggregates with a diameter (d, mm): 0.25 > d > 0.05; *R* is the proportion of ESP of the same dimensions.

A wavelength-dispersion X-ray fluorescence spectrometer was used to determine the content of macro- and microelements (essential group). Colors (dry moist) were described using the Munsell-System [25]. The results of cluster analysis (Ward's method, Euclidean distance) were based on the standard deviation normalized values of soil parameters for members of a series of agrogenic soil transformations.

### 2.3. Physical and Chemical Analyzes of Soils Sampling

Results of soil genetic studies for 4 polygons in Moldova and Bessarabia [26], Northern Black Sea Region [27,28], in Steppe Crimea [29], on the territory of the Central Black Earth Region [30] were systematically generalized with the presentation of data in the form of series of agrogenic soil transformations in the coordinates of regional climatic systems. In addition to the author's data, statistically generalized data on the soils of Moldova were involved [31]. The boundaries of the ancient agricultural regions on the Crimean Peninsula are substantiated based on the results of many years of geoarchaeological studies of the geography of antiquity agrarian settlements in the northwestern part [32–34] and in the east of the peninsula [35,36]. The thousand-year history of the first large-scale period of land use in the Lower Bug region is connected with the formation of the rural district of the Greek city-state Olbia, which included over 150 rural settlements [37]. Identification of post-antique fallow land was carried out by sequentially combining a previously prepared vector layer of contours of ancient agriculture with satellite images of different times from

different years: images from Google Earth 2005–2020, Landsat 1985–2020 images, and CORONA 1969 image. In addition, archival aerial photographs of the study area were involved, which we managed to tie and combine with traces of land surveying identified from satellite images. A number of soil agrogenic transformations of each of the 4 key sites included five groups of objects: virgin land; modern-day plough-land (<100 yrs); continuous plough-land (>100 yrs); fallow land in the modern era (n·10 yrs); and post-antique long-term fallow land. The total sample for regions (n = 511) had the following distribution by the amount of data: Moldova and Bessarabia (n = 109); Northern Black Sea Region (n = 98); Steppe Crimea (n = 134); and Central Black Earth Region (n = 170).

*2.4. Statistical Processing of Data on the Corg Content*

Statistical analysis of the data was performed in the R 3.4.4 programming environment for statistical calculation using functions from the standard packets set. The additional packet ggplot2 was applied to generate statistical plots. The goal of the statistical analysis was to determine the differences between the Corg content in the studied testing sites and to evaluate the statistical significance of these differences. The statistical analysis was started from an examination of the data distribution. Using the Shapiro–Wilk test (shapiro.test function) it was decided whether the distribution fits the normality. This test has shown that the distribution of the most of the samples is not a normal one. Therefore, nonparametric methods were used in the further analysis. Inside each of the variants of land use, the testing sites under study were compared by the mean values of the Corg content through Kruskal–Wallis one-way analysis of variance while applying the kruskal.test function. In order to define in which particular pairs of the testing sites there are differences, post hoc comparisons were conducted by means of the Mann–Whitney U test using wilcox.test function. The resulting multiple comparisons of the *p*-value were adjusted using Bonferroni's correction.

## 3. Results and Discussion

*3.1. Territorial Features of the Formation of SOM in the Chernozem Belt (According to the Reference Historical Database)*

The soil pool of the Russian Federation constitutes about 12% of soils on the entire terrestrial globe, so their upper horizons must have accumulated at least 23% of the global stock of soil organic matter [21]. In Russia's territory, of primary importance is the role of the most fertile soils in the world—Chernozems occupying 7% of the land surface uncovered by ice [38], contain 2–8% of organic carbon in horizon A [39]. Meanwhile, in the richest soils containing >10% of humus, its stores in a layer of 0–20 cm can attain a content of >200 t ha$^{-1}$ [40]. In regions with a long agrarian history, similar forms of soil degradation are observable: the loss of nutrients through biological carrying out, humus mineralization, loss of structural stability, decrease in the mesofauna activity, soil exhaustion, wind erosion, etc. [28]. The rates of these processes may be assessed both through comparison of ploughed soils with reference data (virgin lands) and using results of historically the first determinations of the soil organic matter (SOM) content. A considerable mass of the data was first examined for writing the book "*Russkiy chernozem*" (Russian Chernozem, 1883) by the founder of the genetic soil study science Vasily Dokuchaev. This highly esteemed scientist discussed SOM as a component of the soil system, important both for diagnosis of soils and as a source of fertility. His well-known definition of Chernozem as the "king of soils" was due to the accumulative type of profile Corg distribution in these soils. The summer months of 1877 and 1878 were devoted by Dokuchaev to the determination of the Chernozem boundaries in the European part of Russia when he covered in a cart a route of over 10,000 km. Using 250 determinations of humus in the top layer of soils sampled in similar relief conditions (at water catchments) [41], Dokuchaev compiled the first map of isohumic lines of the Chernozem zone in the European part of the Russian Empire (Figure 1). Analyzing this map, he established that in the Chernozem zone there are five sublatitudinal lines of SOM differentiation extended from north-east to

south-west. He also showed the transformation of this index from west to east. Chernozems of the south-west contain 2.3–2.9% of SOC, those of central regions contain 4.1–5.8%, and a maximum content of 5.8–9.3% is recordable beginning with the left bank of the Volga River [42].

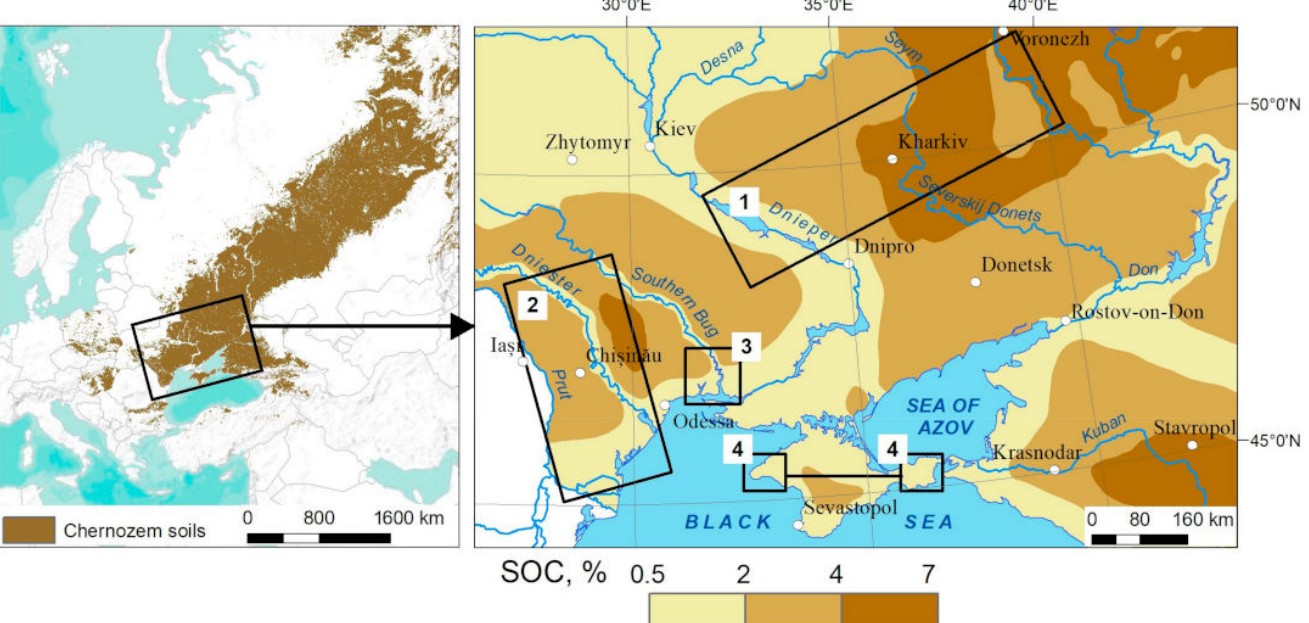

**Figure 1.** Location of study sites in the Chernozem zone in the southwestern of the East European Plain: Central Black Earth Region (1); Moldova and Bessarabia (2); Northern Black Sea Region (3); Steppe Crimea (4). Compiled according to [41,43]. SOM values from map of the Chernozem zone of European Russia [41] were recalculated into SOC values (%).

In order to obtain information on the initial (i.e., in the 1870s) SOC content, we carried out in GIS an assessment of the distribution of areas by three SOC gradations in key regions where the authors conducted soil-genetic investigations in 1985–2023 (Figure 1). Obtained results present more details on those geographic regularities in the SOC distribution which were noted by Dokuchaev. The two first regions (Table 1), where the Chernozem have developed in the forest-steppe zone, possess the outmost areas of soils with the SOC content of 4–7%. Chernozems in the steppe zone, even 145 years ago, did not have a SOC content of >4%.

**Table 1.** Ratio of areas with different SOC content for key sites.

| No. | Region | Area (%) of SOC (%) Gradation | | |
|---|---|---|---|---|
| | | 0.5–2 | 2–4 | 4–7 |
| 1 | Central Black Earth Region | 15 | 41 | 44 |
| 2 | Moldova | 31 | 61 | 7 |
| 3 | Northern Black Sea Region | 58 | 42 | 0 |
| 4 | Steppe Crimea | 100 | 0 | 0 |

Previously, Dokuchaev's map had already been attracted for the assessment of SOM losses over 100 years. Particularly, according to [44] it has been demonstrated that, from 1881 until 1981, Chernozems of the forest-steppe have lost 50–70 t ha$^{-1}$ in a layer 0–50 cm thick, or 30–40% of the SOC initial content.

### 3.2. Climatic Prerequisites for Pedogenesis and the Formation of Humus Profiles in Chernozems

The processes of formation and deposition of SOM to a great extent are determined by climate. The latitudinal gradient of the SOM accumulation process in steppe ecosystems

is characterized by a north-to-south decrease in the thickness of the humus horizon from 130 to 10 cm, a lowering of SOM concentration from 10–12% to 2–3%, and a decrease in its stock from 700 to 100 t ha$^{-1}$ [45].

As a survey of the attempts to link not only simple characteristics of the heat and moisture provision of landscape zones but also many complex indices has shown, these indices poorly reflect the relation of the combination of hydrothermal factors and their limiting combinations with zonal parameters of humus profiles (i.e., their thickness and saturation with Corg). This shortcoming was successfully overcome by designing an integral bioenergetic system of unities (soil and plant paragenetic series) [23], which, in a generalized form, describe the regularities of zonal transformations of landscapes on the Earth's surface.

Within the boundaries of the Chernozem regions studied by us (Figure 2), the values of energy expenses for the pedogenesis bring together regions 1 and 2, where a forest-steppe zone is represented, while two other steppe regions are similar also in terms of heat and moisture provision (Table 2).

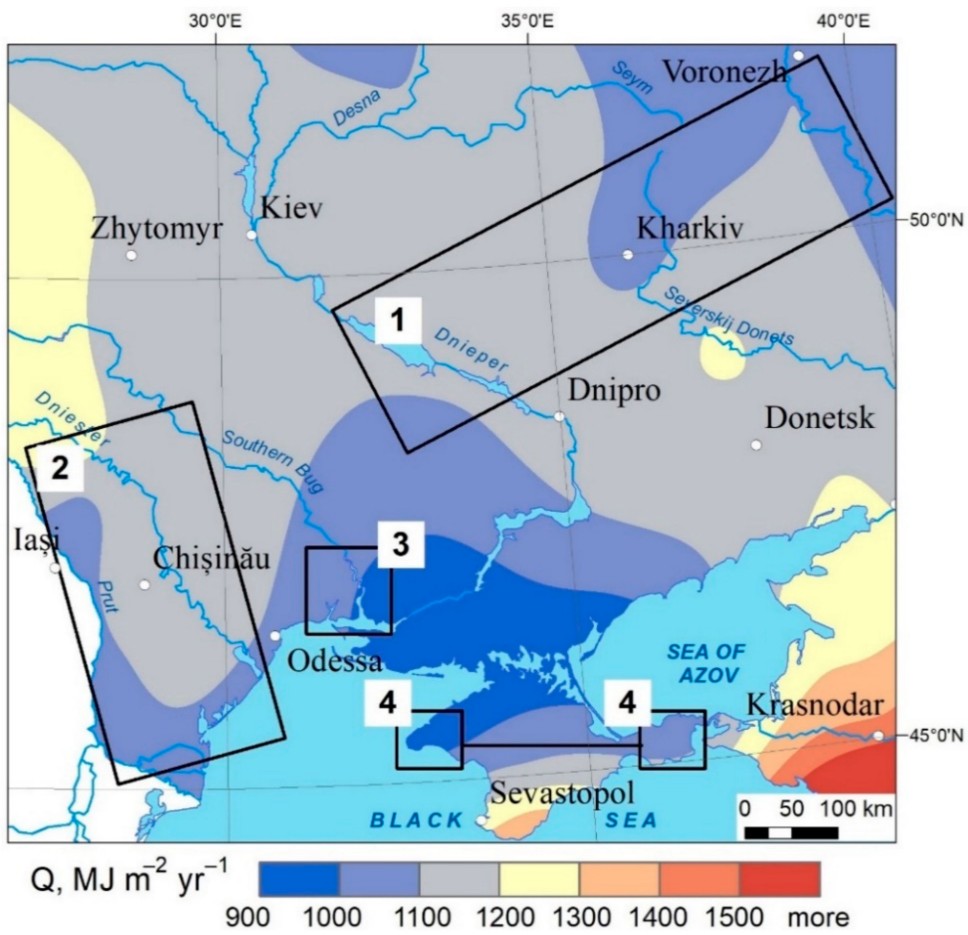

**Figure 2.** Distribution of the energy expenses for the natural process of soil formation ($Q$, MJ m$^{-2}$ yr$^{-1}$): Central Black Earth Region (1); Moldova and Bessarabia (2); Northern Black Sea Region (3); Steppe Crimea (4).

**Table 2.** Average $Q$ values for key sites from 1980 to 2021.

| No. | Region | $P$, mm | $T$, °C | RB, MJ m$^{-2}$ yr$^{-1}$ | $Q$, MJ m$^{-2}$ yr$^{-1}$ |
|---|---|---|---|---|---|
| 1 | Central Black Earth Region | 571 ± 23 | 8.1 ± 0.6 | 1920 ± 68 | 1120 ± 30 |
| 2 | Moldova | 512 ± 46 | 10.3 ± 0.8 | 2188 ± 97 | 1130 ± 45 |
| 3 | Northern Black Sea Region | 435 ± 17 | 10.6 ± 0.3 | 2222 ± 33 | 1014 ± 25 |
| 4 | Steppe Crimea | 426 ± 27 | 11.7 ± 0.3 | 2362 ± 41 | 1017 ± 51 |

The authors have collected data on the facial (provincial) parameters of the top thickness of the humus horizon in the major types of automorphic soils on loose pedogenic rocks in the East European Plain. As exactly as possible, the rock areas were characterized by conditions of heat and moisture provision. There was obtained an equation based on the approximation of empirical data (Figure 3) on the values of the maximum (upper limit) thickness of the humus horizon of soils ($H_{lim}$) in different soil-climatic conditions with the energetic potential $Q$ (2) [46]:

$$H_{lim} = 2000/(1 + \exp(5.346 - 0.0052 \cdot Q)). \tag{5}$$

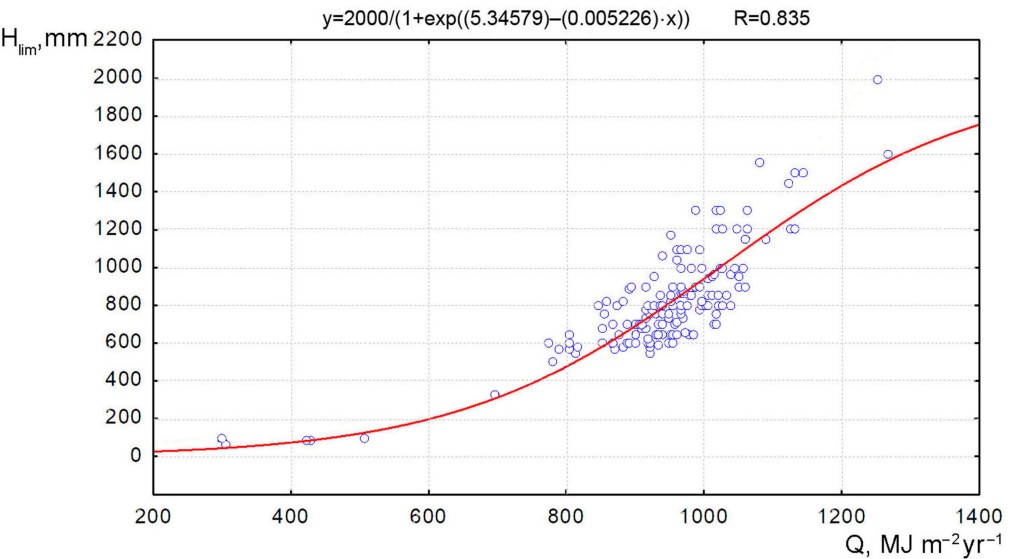

**Figure 3.** Dependence of the maximum thickness of the humus horizon ($H_{lim}$, mm) in zonal soils of the East European Plain on the energy costs for soil formation ($Q$, MJ m$^{-2}$ yr$^{-1}$).

The logistic form of the curve of $H_{lim}$ dependence on climatic energy consumption for soil formation corresponds best to the logic of the process represented in Figure 3. According to this dependence, the vertical Corg differentiation can spread along the profile 200 cm down in the optimal conditions of pedogenesis. The results for such a region so well studied for soil conditions such as Moldova [47] have shown the possibility of describing the vertical SOC differentiation in soils in virgin lands and forest areas by an exponential unilateral distribution while, along the profile of agriculturally long cultivated soils, it is described by a normal unilateral distribution. The type of the SOC distribution diagram for a soil profile is fairly important for the evaluation of the carbon deposition capability of the entire soil thickness. As demonstrated by Figure 1, soils with a humus horizon thickness above 1 m can be generated at a value of $Q > 1000$ MJ m$^{-2}$ yr$^{-1}$.

### 3.3. Statistical Assessment of Differences in the SOC Content in the Regions of the Chernozem Belt and Types of Land Use

An analysis of soil-climatic differences in Northern Eurasia showed that humus formation in the geographical aspect is most definitely detected by the $C_{HA}$:$C_{FA}$ ratio, and among the indicators of humus components, such indicators are the total content of HA (degree of humification) and, in particular, their elemental composition [48]. The dispersion of the SOC content values is shown in Figure 1. For all the variants of land use, similar changes during the transition from one testing site to another are observable. The studied testing sites in Figure 4 are sorted from left to right according to their geographic position (latitude of the locality) from a more northern to a more southern one. In this direction, a decrease in SOC content is observed, which changes into an increase when turning to Steppe Crimea, which, like the westernmost testing site (Moldova and Bessarabia), with its Chernozem zone belongs to the warm Pontic South-European facies in accordance with the

soil-geographic division. The value ranges overlap for all the testing sites. The geographical position of the test sites in the latitudinal direction does not unambiguously determine the differences in the SOC content. In this regard, Steppe Crimea stands out, which, due to the position of the mountains in the south, has an inversion in latitudinal zonality and, for some land use/cover types, is closer to Moldova and the Central Black Earth Region than to the Northern Black Sea Region.

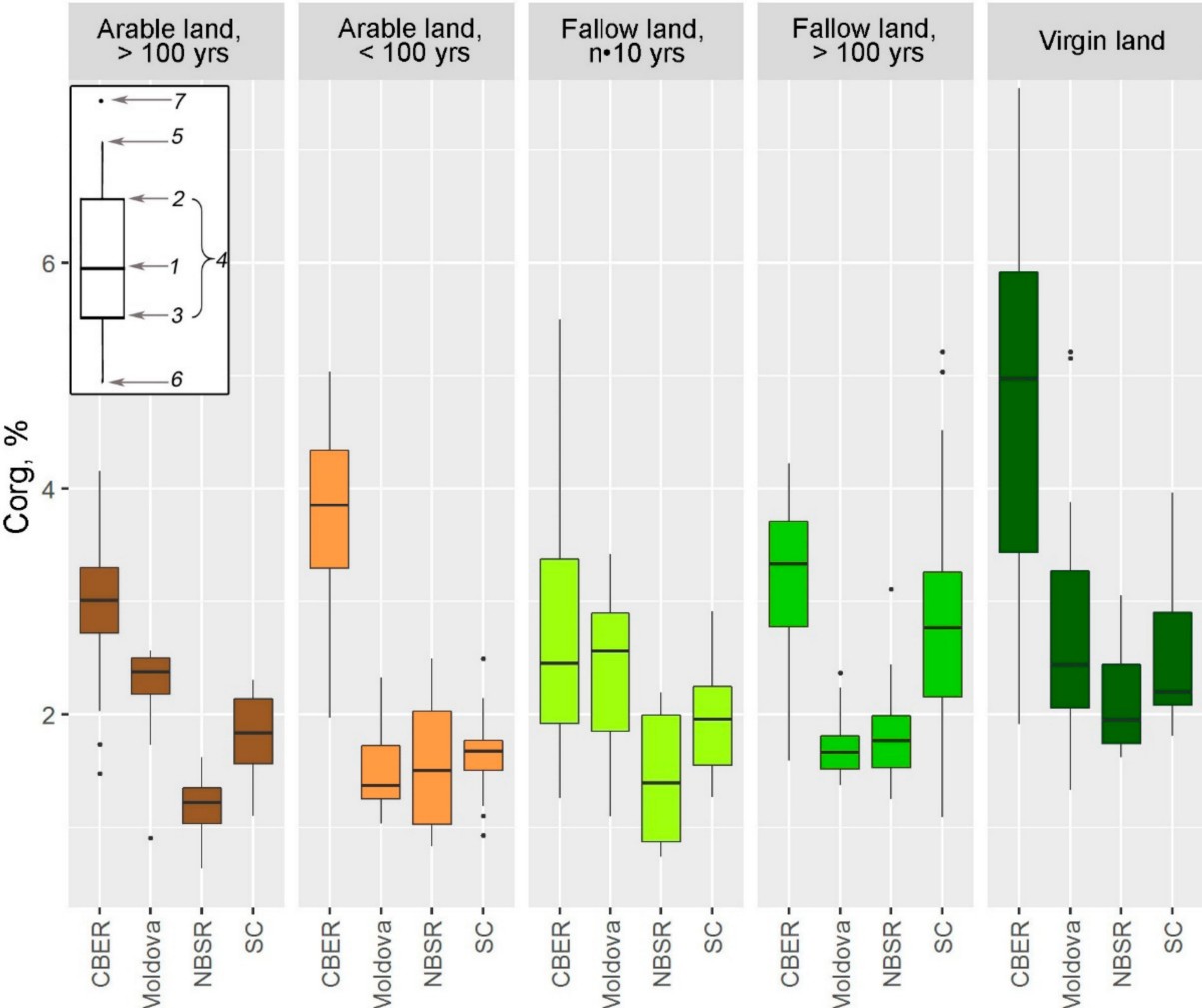

**Figure 4.** Boxplots showing differences in SOC content between landuse/cover types and regions: Central Black Earth Region (CBER), Moldova and Bessarabia, Northern Black Sea Region (NBSR); and Steppe Crimea (SC). For boxplots: 1 is median, 2 is third quartile (Q3), 3 is first quartile (Q1), 4 is interquartile range (IQR), 5 and 6 are maximum and minimum, respectively (excluding outliers), 7 is outliers.

The results of the Kruskal–Wallis test in Table 3 are presented. This test has shown statistically significant results ($p < 0.05$) for all the variants of land use. This fact suggests that belonging to a certain testing site is the key factor influencing the SOC content. For each variant of land use there is at least one pair of testing sites where the SOC content differs.

The results of pairwise comparisons using the Mann–Whitney U-test show that for each of the parties, statistical characteristics of land use, and significant differences are observed in several pairs of comparisons (Table 4). For arable land older than 100 years, statistically significant differences in the content of organic carbon in the soil are observed in all six pairs of comparisons. For a fallow land > 100 yrs, statistically significant differences

in the content of organic carbon in the soil are observed in four pairs of comparisons out of six. For other land use options, significant differences in the content of organic carbon in the soil are observed in three pairs of comparisons out of six.

**Table 3.** Results of one-way analysis of variance according to the Kruskal–Wallis test.

| Land Use | Kruskal–Wallis H Statistic | df | *p*-Value |
|---|---|---|---|
| Arable land, <100 years | 56.30 | 3 | $3.60 \times 10^{-12}$ |
| Arable land, >100 years | 94.50 | 3 | $2.36 \times 10^{-20}$ |
| Fallow land, n·10 years | 24.40 | 3 | $2.02 \times 10^{-5}$ |
| Fallow land, >100 years | 56.10 | 3 | $3.92 \times 10^{-12}$ |
| Virgin land | 41.40 | 3 | $5.28 \times 10^{-9}$ |

**Table 4.** Results of U-test Mann–Whitney (*p*-value) with Bonferroni correction.

| Pairs of Region Comparisons | Land Use | | | | |
|---|---|---|---|---|---|
| | Arable Land, >100 yrs | Arable Land, <100 yrs | Fallow Land, n × 10 yrs | Fallow Land, >100 yrs | Virgin Land |
| Moldova-CBER | $9.64 \times 10^{-9}$ | $4.42 \times 10^{-9}$ | 1.00 | $4.82 \times 10^{-7}$ | 0.0004 |
| NBSR-CBER | $1.31 \times 10^{-11}$ | $3.12 \times 10^{-7}$ | 0.0001 | $6.69 \times 10^{-7}$ | $1.75 \times 10^{-6}$ |
| SC-CBER | $2.19 \times 10^{-8}$ | $7.55 \times 10^{-8}$ | 0.03 | 0.08 | $4.52 \times 10^{-6}$ |
| NBSR-Moldova | $1.03 \times 10^{-8}$ | 1.00 | 0.004 | 1.00 | 0.16 |
| SC-Moldova | $4.77 \times 10^{-5}$ | 0.81 | 0.49 | $2.20 \times 10^{-6}$ | 1.00 |
| SC-NBSR | $1.49 \times 10^{-5}$ | 1.00 | 0.12 | $1.46 \times 10^{-5}$ | 0.11 |

Abbreviations for comparison pairs: Central Black Earth Region (CBER); Northern Black Sea region (NBSR); Steppe Crimea (SC). Green background: statistically significant differences.

Thus, the agrogenic evolution of soils, which is reflected in the characteristic parameters of the SOC content for arable land with a farming duration of >100 years, leads to the individualization of certain regions of the Chernozem zone in the south of the East European Plain (Moldova, southern Ukraine, southwestern Russia). The specificity of the humus state of the regions is mainly preserved during the long-term fallow regime of land (>100 years), although when comparing the western regions (Moldova with Bessarabia and the Northern Black Sea Region) and regions with a temperate continental climate (Central Black Earth Region and Steppe Crimea, where the average SOC values are 3.2% and 2.8%, respectively) in this respect statistically insignificant differences in the SOC content were established.

The traditional view on virgin land as a reference in consideration of the formation of many soil properties and, in particular, of the percentage and stock of Corg, in some cases may be corrected as it is distinctly demonstrated in Figure 5 concerning the ratio of post-antique long-term fallow land in the region of Steppe Crimea, and a tendency for the Northern Black Sea coast. This phenomenon has already been explained in detail through the results of investigation of soils in the rural surroundings of Ancient Olbia, which, after the large-scale land cultivation at the end of the Archaic period (5th century BC) and almost a millennium (with some interruptions) of arable regime, for over 20 centuries had remained fallow [28]. The rotation regime of land use (from ploughed field to fallow and vice versa), which was traditional in ancient agriculture, involved effective mechanisms of reproduction of soil fertility. Soil, as a self-organized system, can achieve the most thermodynamically stable functioning through alternation of its conditions in the regimes of land use. This process was combined with certain degradation of some properties in the progradation, i.e., the directed renaturation at the postagrogenic stage of evolution. E.g., it has been shown that if the share of labile carbon among its total amount in virgin land constituted 2.5–3.2%, then in soils of the post-antique fallows it was 5.6% and in long-ploughed soils, its content was 6.0–7.3% [28]. In two ancient agricultural regions (Northern Black Sea Region and Steppe Crimea) post-antique long-term fallow land is enriched 1.3–1.4 times more in the Corg content than fallows in the modern era (n × 10 yrs).

Post-antique long-term fallow lands in Steppe Crimea have a higher Corg content than their virgin analogues.

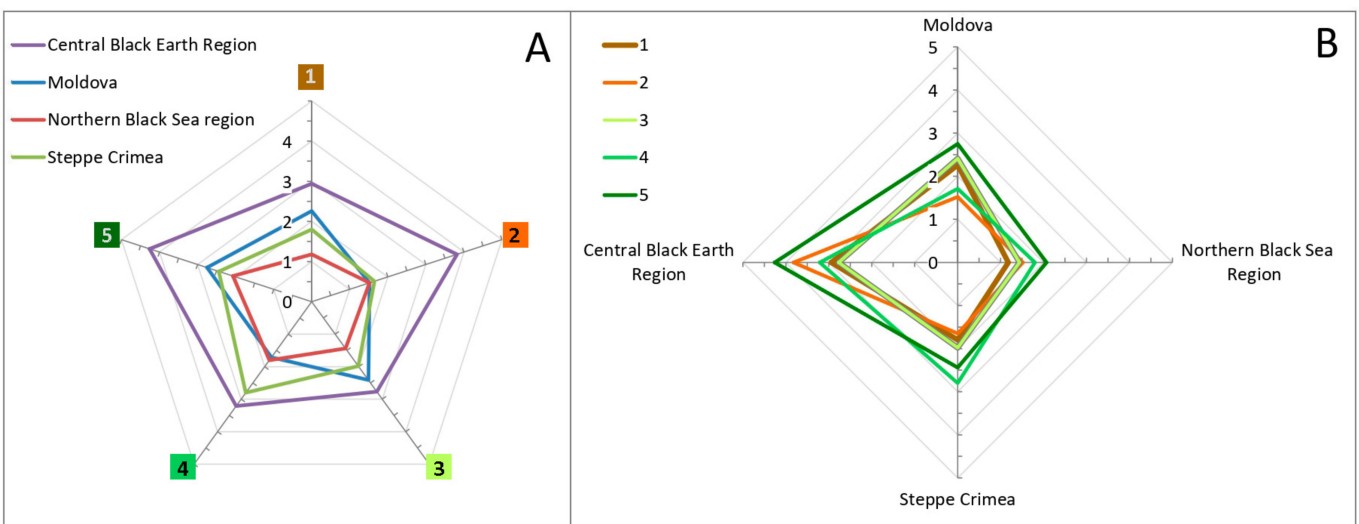

**Figure 5.** Comparison of SOC (%) content by key sites (**A**) and by land use types (**B**): arable land > 100 years old (1); arable land < 100 years old (2); deposit n·10 years (3); fallow land > 100 years (4); virgin land (5).

Ploughed soils of long-term cultivation and recently developed lands little differ in the two regions: Northern Black Sea littoral and Steppe Crimea. Depending on the duration of soil cultivation, the most considerable losses of Corg (by 20 relative %) have been recorded for soils enriched with SOM in the Central Black Earth Region. Arable soils in Moldova are distinguished in the fact that at the new stage of development they have 1.5 times poorer Corg content than long-ploughed soils. Evidently, the effect of progradation in this region with its specific bioclimatic conditions is also observed. Moreover, it should be noted that the history of agriculture among the tribes that inhabited the area within the borders of what is now Romania and Moldova started extremely early, i.e., 8000 BP.

During the realization of field investigations, the method of series of agrogenic transformations of soils was the main technique of studies. This method implies the analysis of differences in the Corg content through comparison of a virgin land with ploughed fields of different terms of development and cultivation based on the principle of a single difference. Nevertheless, through the integration of the set of full-term series of agrogenic transformations of soils at the latitudinal-provincial aspect of the Chernozem geography, it is also possible to reveal stable regularities of the transformation of Corg in particular bioclimatic conditions.

Owing to the fact that the Central Black Earth Region in the forest-steppe and steppe zones is situated, Chernozems predominate here. The part of this region characterized by especially favorable bioclimatic conditions is occupied by thick Chernozems Chernic (typical) containing up to 5.2–5.5% of Corg, while in the southern forest-steppe zone, Chernozems ordinary are represented with Corg content equal to 4.1–4.9%. With such high stores of SOC, even the agricultural loads of 200–300 years have defined a peculiar position of these soils in the coordinates of an agrogenic series, as demonstrated by Figure 5. The virgin land and ploughed lands of different periods preserve a stable regularity in a decrease in Corg percentage from Moldova towards Steppe Crimea and the Northern Black Sea Region; however, as fallow lands of different periods in Steppe Crimea are concerned, the regeneration potential is here more significant and, in the case of post-antique long-term fallows, it is comparable (or statistically close) to soils of the Central Black Earth Region.

### 3.4. Key Features of the SOM Degradation Process as a Result of Plowing

The dynamics of SOM loss over a century is presented by studies in the Michael's Virgin Land Nature Reserve (Ukraine (Sumy Oblast, value $Q = 1050$ MJ m$^{-2}$ yr$^{-1}$)), where in the 0–12 cm layer of Chernozem typical after plowing virgin soil (9.4% SOM) in the meadow steppe after 12, 37, 52, and 100 years, were lost SOM 17, 23, 59, and 63%, respectively [44]. The data indicate that the time dependence of SOM losses has a logarithmic form and after 60–80 years of soil cultivation the rates of SOM loss slow down.

Over the entire history of agricultural development of soils in the south of the Central Russian Upland (average age of tillage 240 yrs), 24% of the initial stock of humus has been lost in one-meter thickness of Chernozems while the annual decrease in the mass of organic matter is evaluated as 540 kg ha$^{-1}$ [48]. The losses of organic matter of Chernozems in the center of the East European Plain resulting from 'ploughing out' (not counting the erosion) have been continuing during the entire period (centuries) of the continuous agricultural development of the territory. Moreover, as it is reflected in Figure 6, the intensity of the humus loss after 50–70 years of ploughing of Chernozems noticeably decreases [49].

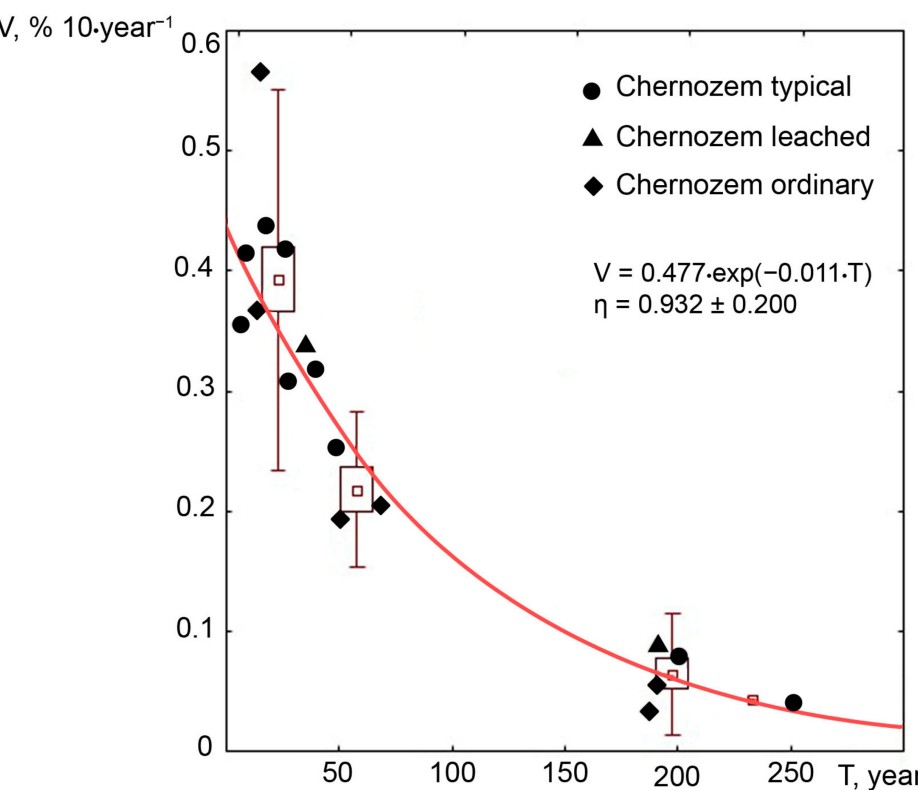

**Figure 6.** The rate (*V*) of decrease in the absolute Corg content in the arable horizon (layer 0–25 cm) during the plowing of non-eroded heavy textured Chernozems of the East European Plain (Adopted from [49]).

It is believed [50] that the major emission of C-CO$_2$ from soils into the atmosphere due to the mineralization of stable groups of humic substances (first of all, the content of HA) is determined mainly by the upper (0–20 cm) humus horizons. Nevertheless, the stock of SOM (t ha$^{-1}$), which is a more unbiased indicator than the percentage of SOM, accounting also for the density of the formation along the profile, has shown in the example of Chernozem typical [51], p. 70, that, with the increase in cultivation duration, SOM loss is occurring differently in the upper and lower half-meter stratum of the soil profile. The Chernozems typical under tillage, by contrast to the reference soils (virgin land where 541 t ha$^{-1}$ in one-meter layer can be considered as one unit), lose the stores of SOM in a layer of 0–100 cm regularly for 8, 22 and 67 years in the proportions: 0.91:0.88:0.73, respectively. Moreover, it is noteworthy that by 67 years of ploughing, a 50–100 cm layer is

subjected to a loss by eight relative percent greater than the layer 0–50 cm deep; however, the more considerable rates of SOM loss during the first two decades are noticeable in a layer 0–50 cm thick (Figure 7).

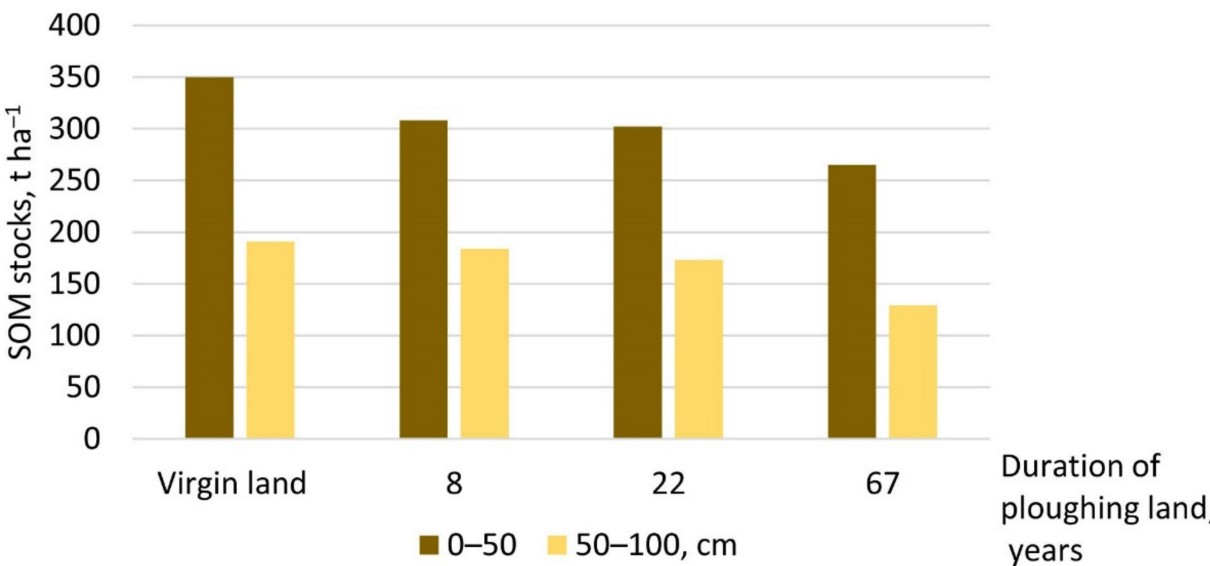

**Figure 7.** Changes in humus stock in soil layers 0–50 cm and 50–100 cm with different durations of ploughing practice land use: (1) virgin land, (2) ploughing lands of different ages (Adaptation from Table data from [51], p. 70).

We converted into a diagram the table of summarized data on the change of humus content and its loss in layers (0–30 cm) of Chernozems in the European part of USSR ploughed for 100 years (1881–1981) [51], p. 69. To the data (as Corg) we also added the values of energy consumption for soil formation (*Q*) (Figure 8).

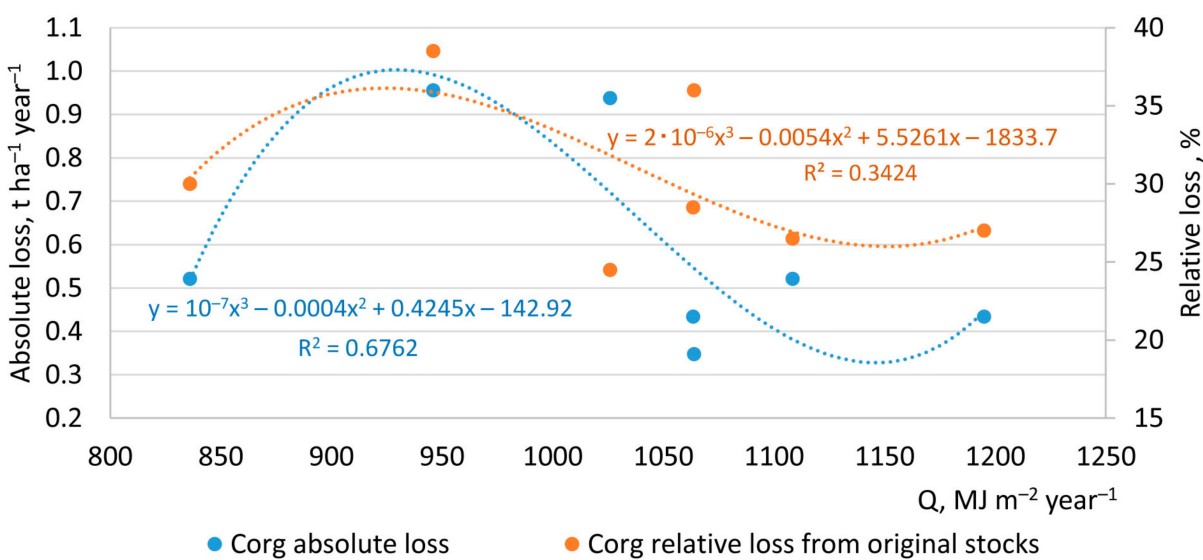

**Figure 8.** Rates of Corg losses (absolute and relative from original stocks) depending on the magnitude of energy costs for soil formation (*Q*, MJ m$^{-2}$ yr$^{-1}$).

Analysis of accumulated Corg stores in the Chernozem zone from the submeridional viewpoint suggests that, despite the general decrease in the annual sum of precipitation from the northern to the southern subtypes (by 220 mm on average) and the natural increase in the evapotranspiration, the maximum of the carbon deposition occurs in the bioclimatic

conditions where there is a balance between precipitation and evapotranspiration. This regularity to some extent is demonstrated also by the form of the dependence of two indicators (annual Corg losses, t ha$^{-1}$, and Corg losses from the initial stores, %) on the climatic energy consumption for soil formation ($Q$) as a 3rd power polynomial (Figure 8). During a century, the total value of Corg loss in the ploughed layer (0–30 cm) in particular subtypes of Chernozem varies within a range of 0.35–0.96 t ha$^{-1}$ per annum amounting, on average, to 0.59 t ha$^{-1}$ per annum (Figure 8). At the same time, there is evidence that while Chernozem typical of the top layer has lost 59% of SOM for 100 yrs, its volume grew by 11% at the depth of 50–60 cm, while at a depth of 140–150 cm its growth amounted to 8% [51], p. 70.

Both the chemical and biological nature of the correlation between the humus content in soil and the gross nitrogen balance is quite clear. At the same time, the ratio between nitrogen and humus equal to 1:20 is a fortuitous value which, as investigations have shown [51], does not remain the same with the transition from one sample to another even within a single taxonomic group of soils. The change of the C:N ratio is related to fluctuations in the relative content of HA; the C:N ratio becomes wider with the increase in their amount. As a result of agrogenesis, the narrowest C:N ratio (7.8–8.0) is acquired by long-ploughed soils and soils ploughed for 100–130 yrs. This implies that in the process of humus loss, carbon-rich HA are lost more rapidly than the fractions of organic substances with a high content of nitrogen of fulvic acids [27]. However, while the annual emission of carbon from soils over entire Russia, according to the data presented by Kudeyarov et al. [52], amounts to 3120.5 million t, by contrast, the share of C-CO$_2$ emission, owing to mineralization of the stable groups of HA (i.e., excluding labile and part of detrital OM [53]), constitutes only from 0.97 to 3.87% of the total emission by soils of Russia [50].

Of the steppe regions which, in addition to the typical problems of present-day agriculture (unbalanced crop rotations, the lack of fertilizers, water erosion, etc.), are influenced by irrigation, a decrease in the percentage of Corg by 14% in 2013–2020 is notable by contrast to the precedent 40-year period [54].

It is notable that the estimates cited above are generalized data on the biochemical losses taken together with mechanical losses and those resulting from solid runoff caused by erosion. The ratio between the rates and volumes of the average humus losses in soils ploughed on slopes and losses on absolutely even surfaces is estimated as equal to 3:1 [49]. If the loss percent is considered relative to the reference soils, then Chernozems with a weak, mean and strong extent of erosion have lost 17, 38 and 60% of SOM, respectively [51].

The process of humus decomposition differs significantly between soils in automorphic positions and those on slopes because of the soil erosion on the latter. This is a selective process since the surface runoff carries out primarily particles with a smaller specific mass and mostly with a diameter < 0.02–0.01 mm enriched with SOM. According to our summarized data, the coefficient of Corg content excess varies from 1.34 (typical) to 1.5 (southern) in the <0.01 mm fraction over its average weighted content in the soil mass for the main subtypes of Chernozem.

Deluvial soils have a humus profile 2–4 m thick with the maximum of accumulated SOM at the boundary between the new deluvium and the buried layer. As soil examinations of land-use areas in Moldova have shown, these soils occupy from 1 to 5% of the area [51]. The formation of pedosediments at the relief secondary elements (trails of the slopes, ravine bottoms, etc.) is not exclusively a result of the mechanical transportation of humic soil from the eroded slopes. It has been demonstrated [55] that, due to SOM input pulsation (caused by differences in the energy potential of showers) followed by its burying them, the repacking of newly deposited SOM occurs, contributing to carbon accumulation in the depositional zone.

*3.5. Soil Carbon Sequestration*

The combined influence of microbial respiration, erosion, or leaching determines the loss of soil capacity to conserve organic carbon soils are to be characterized, i.e., the process

of soil carbon stabilization [56]. As shown above, the study of the influence of mechanisms for C stabilization in the pedosphere is of great importance for the global carbon cycle. Previously, it has been shown [57] that three key mechanisms for SOM stabilization are interacting in the soil system: (i) physical protection, (ii) stabilization by organo-mineral bonding, and (iii) biochemical stabilization. Moreover, of special importance is the biochemical tension of pedogenic processes which is determined by the accessibility of SOM to microbes and enzymes, as well as by the resistance of organic molecules against microbial attack [58].

Soils are to be characterized not only by the process of humus accumulation but also by the ability to form its qualitative composition. On mineral substrates/matrices organomineral complexes are formed with different dimension and aggregation extent and they become dispersed through different granulometric soil fractions [59]. An analysis of the soil carbon sequestration problem cannot be regarded as unbiased if a series of limitations is not taken into account. Among the latter, of note is the soil's propensity to approach saturation levels [60] and the necessity to take into consideration the quality of SOM in order to assess the rate of stabilization.

Investigations of the humus formation using organic substances labelled with $^{14}$C have shown that this process must be considered not as any independent phenomenon but in an inseparable relation with particular bioclimatic conditions [61]. This conclusion is true concerning also the process of deposition of Corg within organo-mineral complexes after humus formation. Thus, in Chernozems, where water migration is limited during the saturation of the root-inhabited layer with SOM, there occurs almost complete mineralization of simple monomer compounds (phenols, amino acids, sugar, etc.), while predominantly the carbon from lignin is included in the humus composition [61]. Intensification of the continental climate character is reflected primarily in the increase in the amplitudes of climatic parameters thus influencing the intra-annual SOM dynamics. An investigation of the seasonable SOM dynamics in the 0–10 cm layer of Chernozem of a protected steppe [62] has shown that, for the time scale of six months (from May to March of the next year), the Corg content fluctuated even 1.6 times (from 4.3% to 6.7%).

The mesofauna, particularly earthworms, has its share in enriching aggregates with organic matter. Thus, according to our data, virgin land in the steppe zone contains from 20 to 55 mass percent of coprolites in some fractions of aggregates (1–10 mm). The coefficient of carbon enrichment of earthworm coprolites defined through the ratio between the Corg content in coprolites and its percentage in macroaggregates of the same diameter is 1.12 for fractions of 3–5 and 5–7 mm; however, in fractions of 2–3 mm no enrichment has been found. The coprolites 2–5 mm in diameter from the parent rock (loess) contain 1.1% Corg, i.e., 1.33 times less than coprolites from the humus horizon.

### 3.6. Evaluation of the Aggregating Efficiency of Corg for Different Types of Land Use

All fractions of a microaggregate composition contain non-aggregated elementary soil particles (ESP) while their ratio to the microaggregates in one or another fraction depends on the features of the parent rock and the type of pedogenesis. In a general form, this ratio is reflected by the structure of the formula for calculation of the microaggregation coefficient (AC). The efficiency of the Corg aggregative capability is distinctly assessed through the ratio AC/Corg, which indicates what degree of aggregativity (in %) is provided by 1% of Corg in a given soil type. Examination of the dependence of AC of a 0–20 cm soil layer on its Corg content reveals no relation of the latter with AC for Chernozems of ploughed fields (Figure 9). This fact, evidently, is caused by the sharp narrowing of the range of Corg variance in the latter (from 1.0 to 2.8%). At the same time, the ploughed lands are characterized by a high extent of microaggregation (AC = 34 ÷ 48%) suggesting a more complicated mechanism of aggregation in which not only SOM takes part.

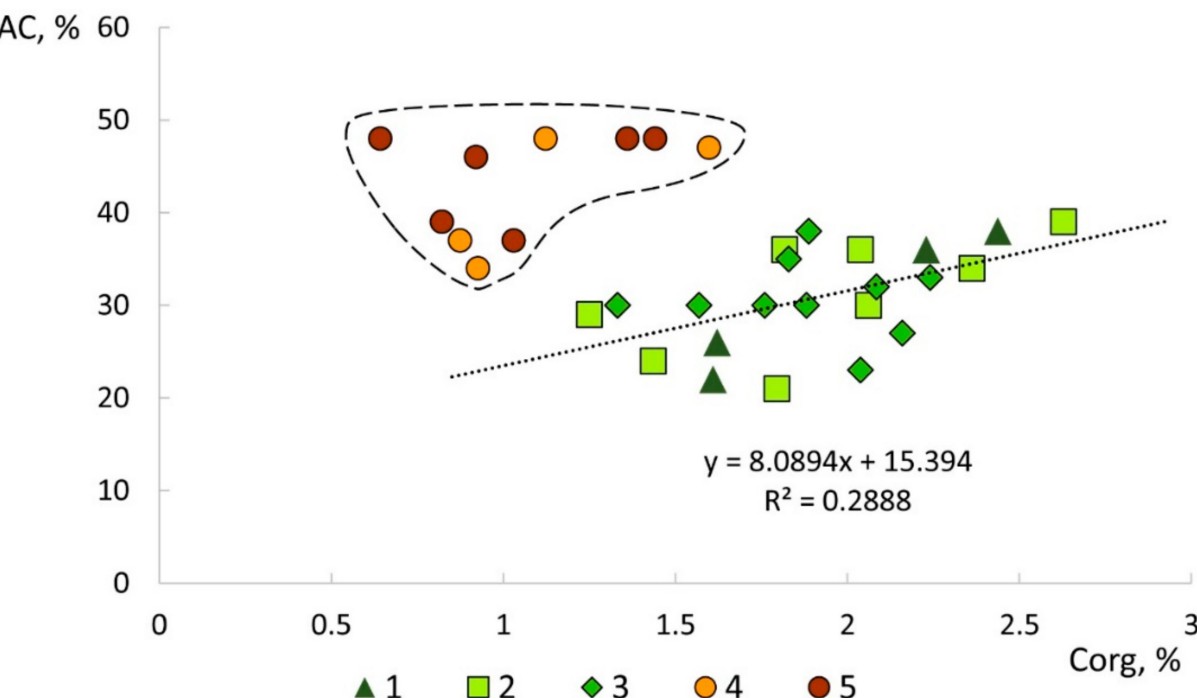

**Figure 9.** Dependence of the microaggregation coefficient (AC) on the Corg content in the 0–20 cm layer of ploughed and uncultivated soils (Northern Black Sea Region, $Q$ value from 800 to 1030 MJ m$^{-2}$ yr$^{-1}$). Soil: virgin land (1); post-residential fallow land (2); long-term abandoned cropland (3); ploughed land in the last 100 years (4); continually ploughed land (>100 yrs) (5).

In intensively cultivated soils, the share of the newly formed Corg in the aggregation process is extremely small because of the lack of plant remains. Hence, the noted level of humus content may be explained by the presence of strong organo-mineral colloids which preserve SOM for a long period. Thus, the microaggregation of ploughed and long-ploughed lands can be affected mainly by the impact of polymeric colloids. The latter, as established by A. Kullman [63], possess aggregation efficiency similar to that of humic substances. For example, among the polymeric structurant for soil protection, it is recommended to use polycomplexes with the participation of polymer nets (hydrogels) based on cross-linked water-soluble cellulose derivatives [64].

In soils formed under natural vegetation (virgin or fallow lands), in the conditions of continuous input of newly formed Corg, the differences in the humus content are reflected in the attained degree of microaggregation (Figure 9). In ploughed soils, a decrease in Corg content is noted by contrast to mature virgin soils and to fallow soils of different time periods. However, the differences in the aggregation efficiency of Corg do not correspond with the acquired depth of humus decomposition, therefore primarily the increase in microaggregation of ploughed soils is of principal importance. Also search for peculiarities of the group and fraction composition of Corg in ploughed soils seems of significance. Analysis of the group and fraction composition of Corg (the content of HA and FA, as well as their 3rd and 4th fractions, respectively) shows that the revealed regularity is strengthened by a fraction of HA, both free and bonded with labile sesquioxides ($Na_2O$, $K_2O$, $CaO$, $MnO$). The percentage of this fraction increases by 1.2–2.3 times in old-ploughed soils as compared with the reference soil (virgin land). Evidently, along with water-soluble humus, this fraction can be of essential importance in the saturation of humic substances with functional groups producing peripheral elements for the formation of heteropolar organo-mineral complexes; the latter, in turn, form 'labile' associates and microaggregates [65,66]. Thus, not only labile fractions of SOM are reproduced more intensively in ploughed soils (in comparison with virgin land), but the SOM aggregation potential is also more completely realized. In this connection, the paradoxical, at first glance, regularity of the decrease in

the sum of non-aggregated ESP with the duration of the soil cultivation (from $23.0 \pm 1.7\%$ in virgin lands to $13.7 \pm 1.6\%$ in a ploughed field of different time periods of cultivation) becomes quite explainable.

### 3.7. Reproduction of SOM with Different Reserves of Plant Matter

In addition to abiotic mechanisms that affect the stabilization of SOC, such as organo-mineral interactions in particular, the second type of mechanisms includes biotic mechanisms that are determined by living soil biomass and soil biodiversity [67].

The regeneration of the humus profile in the conditions of recurrent evolution would be determined by the growing tension of the biological processes such as the transformation of the structure of dead organic matter input (roots, plant litter) and the latter's transformations on the surface and in situ of the soil profile. Pools of detritus are regenerated relatively fast (already in the first years of the fallow regime), as well as pools of weakly bonded (labile) organic matter, since these components of the soil organic substances are the most demanded by the ecosystem as the main reserve of the nutrient substratum for microbe associations. In the plant succession, this period coincides in time with the formation of a layer densely pierced with thin filaceous roots of shrubby (grassy) cereals (fescue, feather grass, June grass, sedges). This process afterwards is slowing down for hundreds of years with the characteristic time of stabilization of the total SOM content at a level of 6–7%. Assessments of carbon sequestration [68] have shown that the storage of carbon in a stable solid form through different mechanisms of its fixation in soil can potentially provide an annual growth rate of 0.4% in the soil carbon stock.

In the situation of a continuous input of anthropogenic energy (land cultivation), the formation of SOM, given a sufficient intake of plant remains, can be limited only by the regeneration of the pools of detritus (mortmass) and labile humus. The formation of these components of the soil organic matter takes place in the porous body of soil, whereas, for reactions of polycondensation of the peripheral humus occurring on the surface of elementary soil particles, a stabilization of the soil mass is thermodynamically more coherent in the structure of the regenerated postagrogenic horizon. As a survey of a large number of modern sources has shown [69], a profound understanding of the changing pool of soil carbon and its fractions provides a scientific basis for minimizing carbon emissions and transforming of the traditional practices of land use with accentuation on carbon neutrality.

### 3.8. Comparative Analysis of the Agrogenic Series of Chernozems with Different Times of Arable and Fallow Regimes

In the buffer strip of investigated land at the transition of the forest-steppe soil zone into the steppe area, thick fertile Chernozems Chernic (ordinary) were found (2 km to north-east from the village of Viktoropol (Veydelevsky District; $Q$ value is 1125 MJ m$^{-2}$ yr$^{-1}$)). Their thickness is 42 cm of horizon A and 82 cm of the humic horizon; a slight effervescence from HCl is observed beginning from the depth of 35 cm and strong effervescence—beginning from 60 cm where carbonate mildew appears in the profile. The SOM content in the top layer reaches 9.5%, and the visible forms of carbonate salts lie in the profile at a depth of 140–160 cm. In terms of the botanic-geographical definition, the territory under study is situated in the steppe zone; it is characterized by small forest masses and steppe gramineous vegetation. Archive materials from the Veydelevka Museum of Regional Studies and sources from the boundary of the 19th–20th centuries examined by us mention that in 1830 Count N. Panin founded here an estate later named Viktoropol. The major arable land masses appeared here 160–170 years ago with the start of cultivation of new agricultural crops: sunflowers, potato plants, and anise, so that by 1858 the share of the ploughed fields made up 65% and the percentage of hayfields was reduced down to 17%. In 1908, when the percentage of ploughed fields in the neighboring farms dropped to a critical level (72–75%), while in the peasants' economies, where even slopes of ravines were ploughed, it exceeded an acceptable rate (84–90%), in S. Panina's estate (now the *Urochishche* Gniloye), a reserve

of the virgin steppe was established measuring 12 *desyatinas* (one *desyatina* = 10,900 m$^2$ or 2.7 acres) and by 1914 its area had reached 50 *desyatinas* (54 ha) [70]. This was one of the first preserves founded in that time. In the same 1908, according to V. Dokuchaev's design, the single in Russia and the locality meteorological station was here opened with the assistance of S. Panina. There the vegetation of the untouched virgin steppe was studied and, in particular, the features of plant processes, and their interrelations with the soil and climate. In 1910–1914, part of the lands of Viktoropol (the northern section of the *Urochishche* Gniloye) was allotted for planting a broad-leaved forest.

The narrow-leaved Stipa steppes in Belgorod Oblast were first studied by the botanist B. Keller in 1931 near the village of Veydelevka where they received the name of the Veydelevka steppes. The narrow-leaved feather-grass steppes explored by Keller are preserved in the vicinity of the villages of Solontsy and Viktoropol in four *urochishches* (isolated tracts of land) within an area of approximately 250 ha. In 1908–1917, the 'Preserve of Virgin Steppe named after Countess S. V. Panina' in the surroundings of Viktoropol (Gorenkov Yar) was functioning within the area of 50 *desyatinas* [70]. Now, the *Urochishche* Gniloye is a regional preserve with the a total area of 60 ha (10 ha—the area of the forest terrain and 50 ha—the area of the steppe tract of Gorenkov Yar). The composition of the steppe associations of the Veydelevsky District includes, although with a low continuity, some typical species of meadow steppes. On the other hand, a noticeable role in the cenoflora is played by species (*Eryngium campestre*, *Phlomis pungens*, *Galium octonarium*, *Salvia tesquicola*) that indicate the belonging of these phytocenoses to the real rich herb-bunchgrass steppes [71]. The protected status of the territory is largely associated with the abundance of Peony (*Paeonia Tenuifolia* L.), which is included in the Red Book of Russia (1984, the status is a rare species) and Appendix 1 of the Berne Convention (2002) (Figure 10, F2 and callout). In 2012, the sites *Gniloye* and *Kamenya* with an area of 220 ha were entered in the list of the Emerald Network of Europe in Belgorod Oblast.

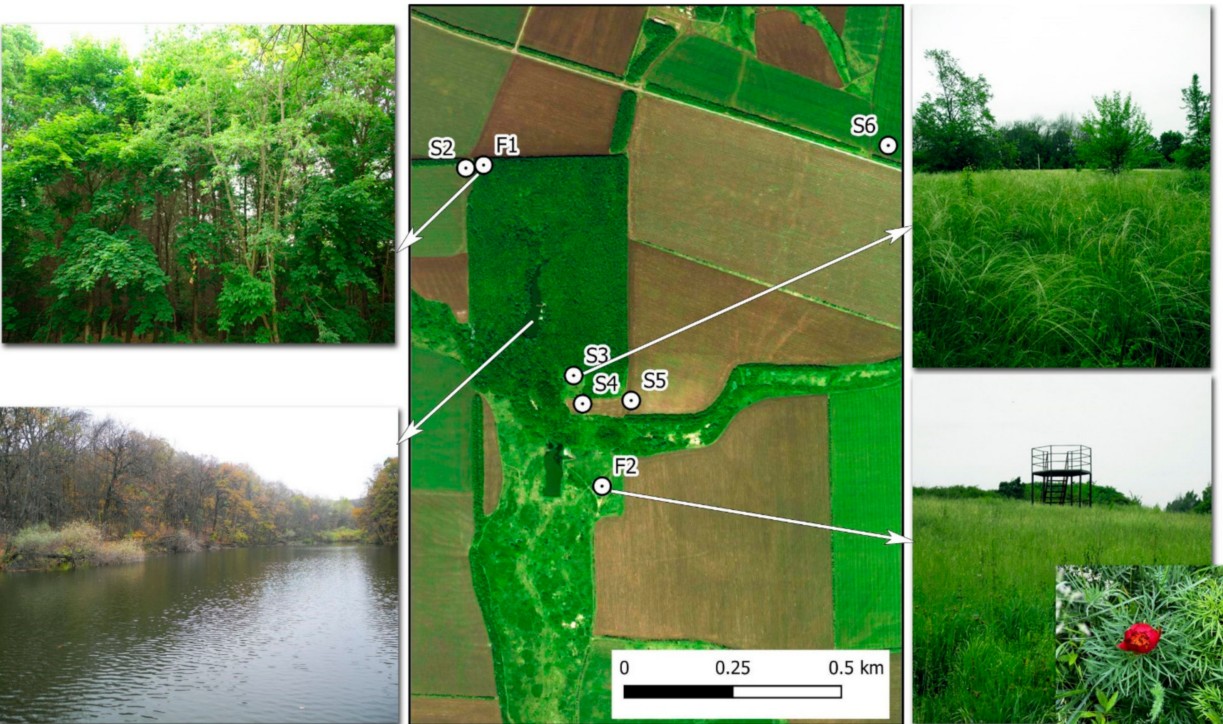

**Figure 10.** Research sites within a regional preserve "Urochishche Gniloye". Designations: arable lands (S2, S6: 60 years; S5: 160–170 years); fallow lands of different times (S3, S4); virgin land (F2); forest of artificial origin (100 years) (F1).

In the course of our own field explorations, an agrogenic soil series in seven areas of different duration and patterns of land use was selected (Figure 10). The investigations comprised: fields tilled at different times (S2, S6: 60 years; S5: 160–170 years), fallow lands of different times (S3, S4), and, for comparison, a virgin land (F2) and a broad-leaved forest of artificial origin (100 years) (F1).

The assessment of the soil quality (SQ) was conducted using the geometric average through 16 independent indices. The fallow lands (S3, S4) have the least values of SQ. This fact allows us to turn to the subject of the interconnection of the notions of soil fertility and soil humic content. The initial soil matrix, i.e., the granulometric composition and mineralogical makeup greatly determine the resources of the soil fertility so that such features as belong to fallow lands (S3, S4) become of priority for the formation of diversity of physicochemical properties reflected in the integral evaluation of fertility. Humus is a very dynamic component of soil, which, by contrast to the other biogenic components of the Earth's crust, not only is accumulating or lost, but is altogether in constant rotation [51].

The results of cluster analysis based on an aggregate of 17 indices of the humic, agrophysical, and geochemical (through the content of seven indispensable macro- and microelements [72]) conditions (Table 5) showed (Figure 11) that, at a high threshold distance, horizons of renaturation are distinguishable at fallow lands (S3, S4), contrasting, as compared with those of ploughed lands of different periods (S5, S6), both in the features of SOM (a 1.8 times higher content of the labile humus and a 1.5 times higher share of $C_{FA}$) and in indices of agrophysical condition (2.5 times higher values of the average weighted diameter of water-stable aggregates and a 1.6 times greater structural coefficient). To a lesser extent, they differ from the ploughed field situated within a zone of influence (70 m) of a woodland (S2); this fact is mostly related to a higher (2.7 times) humic content in such soil with labile humus.

**Table 5.** Basic physical and chemical properties of Chernozem soils (Veydelevsky polygon).

| Indices | Units | S2 | S3 | S4 | S5 | S6 |
|---|---|---|---|---|---|---|
| Layer | cm | 0–16 | 0–13 | 0–14 | 0–16 | 0–14 |
| Corg | % | 3.35 | 3.09 | 1.83 | 3.09 | 3.02 |
| N total | % | 0.33 | 0.28 | 0.17 | 0.29 | 0.36 |
| C:N | – | 10 | 11 | 11 | 11 | 8 |
| Labile humus | % | 0.90 | 0.33 | 0.33 | 0.15 | 0.21 |
| $C_{HA}$ | % | 47.4 | 44.4 | 54.3 | 54 | 66.7 |
| $C_{FA}$ | % | 17.5 | 23.2 | 22.9 | 12.5 | 17.6 |
| CEC | cmol(+)kg$^{-1}$ | 39.7 | 23.0 | 15.0 | 32.8 | 38.6 |
| <0.05 mm | % | 51.24 | 53.15 | 48.94 | 63.74 | 97.52 |
| <0.01 mm | % | 50.82 | 32.96 | 29.56 | 58.24 | 62.75 |
| Cstr | – | 1.12 | 1.85 | 0.80 | 0.75 | 0.89 |
| d | mm | 0.42 | 1.95 | 0.45 | 0.44 | 0.51 |
| $P_2O_5$ | % | 0.16 | 0.09 | 0.12 | 0.14 | 0.16 |
| $K_2O$ | % | 1.94 | 0.96 | 1.62 | 1.76 | 1.99 |
| MgO | % | 0.83 | 0.22 | 0.38 | 0.86 | 0.87 |
| MnO | % | 0.09 | 0.05 | 0.07 | 0.08 | 0.09 |
| $Fe_2O_3$ | % | 4.83 | 2.41 | 3.08 | 4.38 | 4.99 |
| Zn | mg·kg$^{-1}$ | 83.93 | 48.13 | 50.33 | 86.79 | 85.46 |
| Ni | mg·kg$^{-1}$ | 55.09 | 26.41 | 34.83 | 50.94 | 56.25 |
| SQ | – | 3.6 | 2.6 | 2.4 | 2.9 | 3.4 |

The top layers of soils of the agrogenic series differ only insignificantly from each other in the Corg content (3.0–3.4%) except for a young deposit (S4) with some lower content (1.8%). However, essential differences are observable during the analysis of the qualitative humus composition. Only S6 soil is characterized by a high nitrogen richness of the humus among the other objects (C:N = 8); this soil, in addition, possesses an ultrahigh extent of the SOM humus saturation. The fallow lands (S3, S4) are distinguished by a higher content of the labile part of SOM as compared with the ploughed

fields (S5, S6). Meanwhile, this content is fairly important for the formation of agronomy valuable aggregates. In addition, the fallow lands are characterized by a higher content of the carbon of fulvic acids, which are the lightest fraction of the humic substances and possess a considerable biological activity.

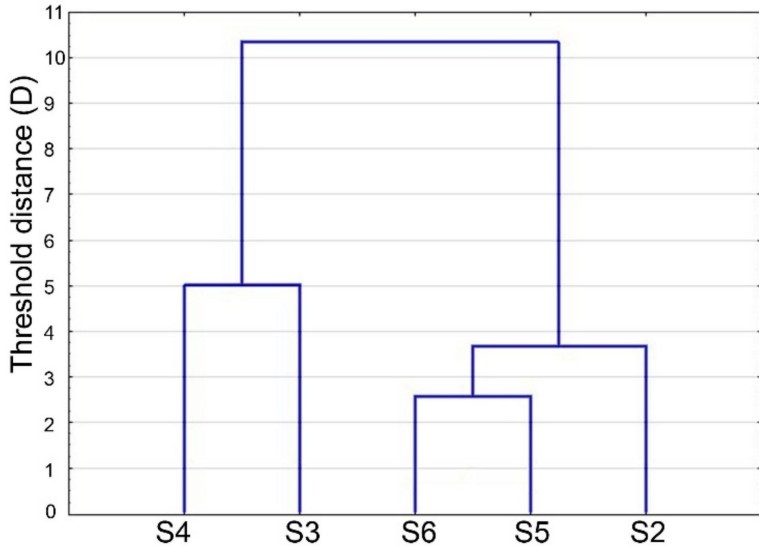

**Figure 11.** The results of cluster analysis according to the Euclidean distance: S2–S6 is members of the agrogenic series of soils.

Some previous studies of seemingly regular changes in such parameters of soil carbon sequestration across a chronosequence as TC and TN fractions of SOM have shown that the restoration process is not always strictly deterministic over time [73]. In particular, the soil of a younger fallow (S4) differs from the soil of a fairly mature fallow (S3) in a higher proportion of $C_{HA}$ (1.22 times), which characterizes a higher degree of SOM humification.

Sediments, delivered from arable lands by surface runoff, are the main source of Corg accumulation in ponds and reservoirs [74], although the influence of processes from the internal life of water bodies cannot be excluded. At our site, the pond in the *Gniloe* tract, which was put into operation in 1910, was characterized by a change in the Corg content in bottom sediments from 4.94% (near the dam) and 6.96% (in the middle) to 7.96% (in the upper reaches), while in the catchment area the content on arable land was significantly less than 4.57% [75].

The assessment of the carbon-depositing potential of soils of different genesis and soil use patterns is a key question in the solution of the task of regulation of the carbon sequestration by soil. The volume of SOM, which is stabilized and protected against decay, characterizes the carbon-protection capacity (CPC) of soil. In addition to the granulometric potential, the role of the mineralogical composition of the soil and, especially, the colloid-mineral fraction is important in the fixation of SOM. As shown earlier [28], agropedogenesis stimulates intrasoil weathering of feldspars (plagioclases and potassium-sodium spars), while the content of minerals of the montmorillonite group (smectite) increases, and this process, along with changes in the group and fractional composition of humus, reveals the reason for the increase in microaggregation in arable soils of different ages. The physicochemical interactions of SOM with mineral soil particles and, in particular, a high reserve of particles of silt (0.002–0.05 mm) and clay (<0.002 mm) in soil are a mechanism of the most reliable stabilization of carbon and its long conservation for centuries [19]. Therefore, the Corg content in the granulometric fractions of silt and clay with a particle size of <0.02 mm can be considered as a measure of CPC [76], as well as the share of particles < 0.05 mm in size, as it was proved in a later work [77]. Calculations [19] conducted using the method of [76] show that in Belgorod Oblast, the Chernic Chernozems (ordinary and typical) with the content 93% and 95%

of the fraction of particles < 0.05 mm have the values of CPC values equal to 30 and 28 g C kg$^{-1}$ of soil, respectively. As the data in Table 4 show, only single studied objects can correspond to the specified values of CPC values in terms of the share of particles < 0.05 mm, i.e., S6 (the fallow land tilled 60 years ago) and the lower section of horizon A (16–28 cm) at S5, as well as the upper 10 cm of the woodland soil near S2.

It was previously demonstrated that the Corg content in granulodensimetric fractions of water-stable aggregates (1–3 mm) of Chernozem typical, in particles measuring <1 μm, is greater than in coarser fractions [42]. Since the granulometric composition differs in two layers of horizon A (24–31 cm) in the content of particles < 1 μm, particularly at the fallow land, we calculated the mean average content of mechanical elements <2 μm (fine silt and clay) in horizon A. In the content (percent) of particles measuring <2 μm, the objects under study constitute the following ranged diminishing series: S6 (70) > S5 (64) > S3 (53) > S2 (52) > S4 (33). On the one hand, the abundance of finely dispersed particles provides a potential possibility of the fixation of humic substances but, on the other hand, it reflects an agrarian load of various duration and intensity. This fact is to some extent corroborated by the position of the studied objects in the ranged diminishing series built according to the values of the weighted average diameters of water-stable aggregates: S3 > S6 > S4 > S5 > S2. The first three terms of this series are the fallow land or the soil, which several decades ago was subjected to this regime (S6).

The ratio of the specific rates of processes of the plant substance formation and destruction in many respects is controlled by particular bioclimatic conditions. The higher the production and lower the rate of destruction, the greater volumes of mortmass are accumulated in the over ground layer. These substances, but more particularly the mortmass of the underground layer, represent a significant reserve for the formation and subsequent deposition of SOC. The share of detritus in a soil layer of 0–20 cm beneath the feather-grass/fescue association usually constitutes 27–32% of the underground phytomass; however, when steppe fires occur, this value increases up to 45%. According to evidence of many years, due to the decomposition of the overground mass and annual fall-off of roots, the functioning of a virgin land with different ratio of steppe graminoids is able to provide 0.98–1.45 t ha$^{-1}$ of Corg per annum in a layer 0–20 cm thick.

As the studies at the testing site of Veydelevka have shown (Table 6), at the beginning of summertime, the mortmass content is still inconsiderable—14% of the soil mass in the 0–12 cm layer (sod horizon) in a virgin land and 8% at a fallow land. However, this detritus contains 18–23% Corg. This is the period of consumption of the newly formed Corg, which lasts with a rapid increase in biomass, which distinguishes this period from the cold season when there is a cumulative increase in SOM [62].

**Table 6.** Mortmass and carbon mortmass content in soils of virgin land and fallow land (June 2023).

| Land Cover (Plant Association) | Layer, cm | Color (Dry) | Mortmass Content, g * | | Mortmass Carbon, % |
|---|---|---|---|---|---|
| | | | DW$_1$ | DW$_2$ | |
| Virgin land (herbs), S3 | 0–6 | 10 YR 3/2 | 10.71 | 12.52 | 18.08 |
| | 6–12 | 10 YR 3/2.5 | 1.46 | 1.57 | 21.33 |
| Fallow land (feather grass), F2 | 0–6 | 10 YR 3/1.5 | 5.58 | 5.97 | 23.51 |
| | 6–12 | 10 YR 3/2 | 1.70 | 1.76 | 21.55 |

* Mortmass content of absolutely dry matter per 100 g of soil: air-dry (DW$_1$); absolutely dry (DW$_2$).

The total stock of C in the ecosystem of oak forests, taking into account the stock of C in tree detritus, forest litter and 0–100 cm soil layer, reaches 262–286 t ha$^{-1}$ at the age of 75 yrs, while 56–89% of the forested area is carbon, which is deposited with a meter layer of soil [78].

## 4. Conclusions

The authors performed GIS analysis of the map of isohumic curves in the Chernozem zone of European Russia established by V. Dokuchaev based on data of 1877–1878. Our analysis has shown that regions with the most favorable conditions for Chernozems formation (climatic energy consumption for soil formation ($Q$) equal to 1120–1130 MJ m$^{-2}$ yr$^{-1}$), where the Chernozems have been developed in the forest-steppe zone, had in the past the greatest areas of soils with SOC content 4–7% (Central Black Earth Region—44%, Moldova—7%). Chernozems in the steppe zone, i.e., in the regions where the southern subtypes of Chernozem are widespread (Northern Black Sea coast and Steppe Crimea), did not have a SOC content > 4% even 145 years ago.

The use of Dokuchaev's map as the first historical collection of data on the SOC content has shown that Chernozems of the forest-steppe (in automorphic positions of the relief) have lost 30–40% of the initial SOC content in a layer of 0–50 cm during the century. The loss of SOM occurs differently in the upper and lower half-meter horizons of the soil profile. Moreover, over the entire history of the agricultural development of forest-steppe soils (the average age of tillage 240 yrs), the losses of SOC in the layer of 0–100 cm have made up 24% of the initial stock.

The slope Chernozems become depleted of organic matter both by the effect of the biological carry-over and because of the soil erosion, which is a selective process that results in the loss of the particles richest in SOM particles < 0.02–0.01 mm in diameter. The coefficient of the excess of Corg content in the fraction of <0.01 mm over its percentage in the soil mass varies from 1.3 to 1.5 in the main subtypes of Chernozems. Chernozems with the highest extent of erosion have lost at least 60% of SOC by contrast to soils on watersheds.

The climatic dependence of pedogenesis is due to the fact that belonging to a particular region represents the key factor influencing the SOC content in the upper soil horizon, as shown by statistical treatment (Kruskal–Wallis test) of over 500 SOC determinations from within the boundaries of the Chernozem zone in the south of the East European Plain (Moldova, South Ukraine, southwestern Russia). At the same time, for each of the five variants of land use (virgin land, arable, and fallow lands of different times) there is at least one pair of regions in which the SOC content differs. The determining role of the duration of agrarian history of land use in the formation of a certain level of SOC content is reflected in the fact that, for an arable land of >100 yrs, the statistically significant values of this parameter are notable in all pairs of comparisons between the southern regions of the East European Plain.

The high carbon deposition potential of soils with a humus layer thickness of >1 m, including Chernozems in optimal bioclimatic conditions, is provided by values of climatic energy consumption for soil formation ($Q$) above 1000 MJ m$^{-2}$ yr$^{-1}$. The climatic dependence of pedogenesis in the Chernozem belt is distinctly reflected in the fact that the decrease in the Corg content at virgin lands and ploughed lands of a different time is observable with the transition from Moldova to Steppe Crimea and to the Northern Black Sea Region. However, the regeneration potential of fallow lands of different periods is more considerable in Steppe Crimea, while at post-antique long-term fallow lands, it is comparable (statistically proximate) with soils of the Central Black Earth Region.

Steppe ecosystems are distinguished in the fact that predominantly the mortmass of the underground horizon is the major reserve for formation and subsequent deposition of SOC. In particular, the underground mortmass at virgin and fallow lands contains from 18 to 24% of carbon.

Interdisciplinary analysis of the differences in members of a series of agrogenic soil transformations has shown that, by contrast to the ploughed soils, the fallow lands possess a higher content of the labile part of SOM, which in many respects determines the formation of the optimum aggregate composition of the soil. They also have a higher content of carbon of fulvic acids characterized by intensive biological activity.

**Author Contributions:** Conceptualization, F.N.L.; methodology, F.N.L. and P.V.G.; software, Z.A.B. and P.A.U.; validation, O.A.M. and F.N.L.; formal analysis, F.N.L.; investigation, F.N.L., O.A.M. and P.V.G.; data curation, F.N.L.; writing—original draft preparation, F.N.L.; writing—review and editing, Z.A.B.; visualization, Z.A.B.; supervision, F.N.L.; funding acquisition, F.N.L., Z.A.B., O.A.M., P.A.U. and P.V.G. All authors have read and agreed to the published version of the manuscript.

**Funding:** This work was funded by the Russian Science Foundation, project no. 23-17-00169, https://rscf.ru/en/project/23-17-00169/ (accessed on 8 August 2023).

**Data Availability Statement:** The data used in this research work are available upon request from the corresponding authors.

**Conflicts of Interest:** The authors declare no conflict of interest.

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
