# Peer review of "Features of Soil Organic Carbon Transformations in the Southern Area of the East European Plain"

_geosciences, doi:10.3390/geosciences13090278_

Round 1
Reviewer 1 Report
This paper has summarized the soil organic carbon (SOC) content in the upper soil horizon for different land use options within the Chernozem belt of the south of the East European Plain. The features have been established, supported by the complete statistical analyses. The paper is well written and should be interesting to the reader of Geosciences. I would recommend it for publication if the authors resolve my concerns/questions:
1. How equation (2) is obtained? Is it from Ref. 23 or other references? If yes, please provide the reference.
2. In line 293, Figure 1 should be referred to Figure 3?
3. In Figure 3, what are the blue and red curves? Is one of them corresponding to the fitting of eq. (5)? If yes, which is it and what is the other one?
4. In line 312, Figure 1 should be Figure 4?
5. In line 314, “The studies testing sites in Figure 4 are sorted from right to left according to their … from more northern to more southern one.” From Figure 1, it seems to be “from left to right” corresponds to “from more northern to more southern”.
6. In line 320, the claim of “The father the testing sites are distanced from each other, the less their ranges of the SOC content values are overlapped.” seems not obvious in Fig. 4, e.g., for Fallow land, n10yrs.
7. In figure 4, what are the dots below/above the boxes? What do the horizontal and vertical lines crossing the boxes stand for? They should be explained in the caption.
8. In figure 8, what are the fitted parameters for the dotted lines? What are the R values?
9. In figure 9, what is the curve surrounding 4 and 5?
10. In line 764, “… in a virgin land and ca 8% at a fallow land”. The “ca” seems confusing.
11. In Table 5, what does “Color (dry)” mean in the third column? How do read the data there, e.g., 10 YR 3/2?
12. The first paragraph in Conclusions needs to be deleted?
The quality of English Language is high. But some sentences are hard to read, like the one in line 238-239.
Author Response
Dear Reviewer, the authors thank you for your work and valuable comments, which we made the most of in the new version of the Manuscript.
Response to Reviewer #1 Comment
I would recommend it for publication if the authors resolve my concerns/questions:
Reviewer comments |
Action taken comments |
1. How equation (2) is obtained? Is it from Ref. 23 or other references? If yes, please provide the reference. |
Equation (2) was obtained by the Authors by transforming units of measure («Due to the correction of formula (1)») using MathLab. In this regard, the link here is inappropriate.
|
2. In line 293, Figure 1 should be referred to Figure 3? |
We agree with the remark, corrected the link to Figure 3 |
3. In Figure 3, what are the blue and red curves? Is one of them corresponding to the fitting of eq. (5)? If yes, which is it and what is the other one? |
Figure 3 was corrected |
4. In line 312, Figure 1 should be Figure 4? |
The text correctly refers to Figure 1. It demonstrates the spatial dispersion of the SOC content |
5. In line 314, “The studies testing sites in Figure 4 are sorted from right to left according to their … from more northern to more southern one.” From Figure 1, it seems to be “from left to right” corresponds to “from more northern to more southern”. |
We agree with the remark, corrected in the text: The studied testing sites in Figure 4 are sorted from left to right according to their geo-graphic position (latitude of the locality) from more northern to more southern one. |
6. In line 320, the claim of “The father the testing sites are distanced from each other, the less their ranges of the SOC content values are overlapped.” seems not obvious in Fig. 4, e.g., for Fallow land, n10yrs. |
line 331-335: The geographical position of the test sites in the latitudinal direction does not unambiguously determine the differences in the SOC content. In this regard, Steppe Crimea stands out, which, due to the position of the mountains in the south, has an inversion in latitudinal zonality and, for some landuse types, is closer to Moldova and the Central Black Earth Region than to the Northern Black Sea Region.
|
7. In figure 4, what are the dots below/above the boxes? What do the horizontal and vertical lines crossing the boxes stand for? They should be explained in the caption. |
Figure 4 shows Tukey's classic boxplots (box-and-whiskers plots). For boxplots, the thick horizontal line shows the median, the top box shows the third quartile (Q3), the bottom box shows the first quartile (Q1), the height of the box shows the interquartile range (IQR). The top whisker for boxplots shows the maximum value (excluding outliers), the bottom whisker shows the minimum value (excluding outliers), the dots show outliers (values less than Q1-1.5 IQR or greater than Q3+1.5 IQR). In the legend of Figure 4, we have added a sheme to explain the structure of the boxplots. Figure 4. Boxplots showing differences in SOC content between different land uses and regions: Central Black Earth Region (CBER), Moldova and Bessarabia, Northern Black Sea Region (NBSR); and Steppe Crimea (SC). For boxplots: 1 is median, 2 is third quartile (Q3), 3 is first quartile (Q1), 4 is interquartile range (IQR), 5 and 6 are maximum and minimum, respectively (excluding outliers), 7 is outliers.
|
8. In figure 8, what are the fitted parameters for the dotted lines? What are the R values? |
Figure 8 was сorrected |
9. In figure 9, what is the curve surrounding 4 and 5? |
The dotted line around (4) and (5) visually highlights a cluster of land uses that is different from the others and is not described by the linear function shown in the graph. |
10. In line 764, “… in a virgin land and ca 8% at a fallow land”. The “ca” seems confusing. |
Corrected. |
11. In Table 5, what does “Color (dry)” mean in the third column? How do read the data there, e.g., 10 YR 3/2? |
The Munsell scale allows you to determine the color of the soil in a wet and dry state. Here, data are given for a more objective - dry state and a code according to the Munsell Scale. |
12. The first paragraph in Conclusions needs to be deleted? |
Corrected. |
Comments on the Quality of English Language The quality of English Language is high. But some sentences are hard to read, like the one in line 238-239. In this connection, his catchphrase telling that Chernozem, due to its habitus, should be called the ‘king of soils’ relates exactly to the saturation of this soil profile with Corg.
|
Given replacement: His well-known definition of Chernozem as the "king of soils" was due to the accumulative type of profile Corg distribution in these soils. |

Reviewer 2 Report
A review for Geosciences-2555930 entitled “Features of soil organic carbon transformations in the southern area of the East European Plain”
Abstract
I recommend adding more information to obtain a valid abstract to be read in isolation.
Additionally, it would be beneficial for the abstract to commence by outlining the methodology implemented, including a reference to the soil and carbon analyses conducted, the depth of the samples, etc.
Moreover, it would be helpful to include information on the bioclimatic conditions, particularly if the research focused on a specific transect between distinct climate types.
The abstract should contain numerical data, including average carbon level values of the soils examined.
Lines 35--36 “…which stimulated effective mechanisms of reproduction of organic substance. -- Throughout the text, I suggest using an alternative term for "reproduction" when referring to the formation of SOM.
Lines 39-- 41: “It is not justified to consider virgin lands as absolute references for evaluation of the 39 humus conditions since analysis of agrogenic series of Chernozems has corroborated an essential 40 role of the soil organo-mineral matrix in the formation of the carbon protection capacity” -- this sentence cannot be understood without reading the entire article and, in its current form, seems like an unsupported statement, because the abstract does not specify the analytical data that was used to define "soil condition" (for example pH, C/N ratio, aggregate stability, water retention, etc.).
Introduction
Line 70: do not use the full words "soil organic carbon" when the acronym has been entered before (the same in lines 82 and 584.
Line 120: “A quantitative assessment of the contribution of the granulometric potential to hu-mus accumulation;” -- It is unclear what is meant by this phrase, which only appears to focus on the effect of soil texture on its biological capabilities. It is important to note the mineralogical composition of soil, particularly the colloidal mineral fraction, as it has a significant impact on carbon sequestration potential. This varies depending on whether it contains active carbonates, kaolinites, smectites or, most notably, amorphous oxides, which have the highest potential to preserve organic matter.
In general, the introduction presents a simplified treatment of the issue of carbon sequestration in soils. For instance, it frequently mentions the impact of conservation agriculture techniques that result in the reincorporation of crop residues into the soil.
However, it is important to note that in relation to the carbon cycle, the total amount of carbon is often as significant as its quality, meaning how much of it has been converted into humic substances, which differ in their chemical composition from plant biomass. Thus, for instance, in a climatic scenario characterised by increased humidity and temperature, which promotes the mineralisation of organic matter, soils with a low carbon content but predominantly composed of well-matured humic matter would demonstrate greater resilience. Contrarily, soils under agricultural or forestry management which accumulate carbon on the surface in easily biodegradable forms would suffer quicker degradation, thereby releasing their CO2 into the atmosphere.Some of it is said in the line 515, but most of the paper is focused exclusively on studying the total carbon levels of the soils.
Material and methods
Line 153: “2.2. Physical and chemical analyzes of soils”
I suggest providing additional information regarding the methodologies employed, particularly if they are not common and utilised globally.
Specifically, it would be helpful to state the specific extractants used for the extraction of humic substances (e.g. pyrophosphate, sodium hydroxide, etc.).
Additionally, it is necessary to indicate the pH value at which the cation exchange capacity was assessed.
It is apparent that while the abbreviation "HA" is accurately entered on line 158, the complete terms are utilized several times in the entire manuscript.
It should be noted that this section does not refer to soil pH determination or carbonate content. Although these can be approximately inferred given that the soils are of the Chernozen kind, there may be significant differences between various regions.
Line 177: To accurately describe the construction of the dendrograms, it is essential to mention not only the clustering algorithm, but also the dissimilarity index utilised, such as Euclidean distance, Mahalanobis distance, etc.
Results and discussion
Line 228: Use SI units (Mg instead of t)
Line 218 “…the nutrients loss through biological carrying out, humus mineralization and its ‘burning out’ during layer turnover, destructurization, decrease of the mesofauna activity, soil exhaustion, deflation, etc” -- I would rewrite or delete much of the text in this sentence: some terms do not seem very familiar or are confusing, others may not be spelled correctly.
Line 236: In the Anglo-Saxon bibliography I see that it is more frequent to refer to Dokuchaev's given name as Vasili or Vasily.
Line 238: “should be called the ‘king of soils’ relates exactly to the saturation of this soil profile with Corg.“ -- I am unclear on the definition of "carbon saturation" mentioned in this sentence. It could potentially relate to the isohumic characteristic of Chernozems, whereby carbon content is spread evenly across the profile. However, the term "saturation" implies a maximum concentration of Corg, which is not applicable to Chernozems.
Lines 246 and 247 (and throughout the text, figure captions…): Ensure consistency in the capitalisation of cardinal points' names.
Line 331 (Table 1) -- Improve the title of Table 3 to indicate what the different numbers in the Table correspond to (P values?)
Line 354: I would change “is able to realize” by “is capable of achieving”, “can achieve” or something like that.
Figure 5 B: capitalize the word "region" on the far right of the graph. Indicate somewhere that the carbon content is expressed in % (or better, in g·100 g-1)
Line 415 and below; “It is believed [49] that the major emission of C-CО2 from soils into atmosphere due to mineralization of stable groups of humic substances is determined mainly by the upper (0–20 cm) humus horizons.” -- I recommend revising the sentence to clarify the meaning of "stable groups of organic matter" and why they are being referenced instead of "non-stable groups" which are easily biodegradable.
Line 426: Stratum or horizon/layer (?)
Fig. 6: (minor) Be consistent in the use of the sign "times" (×) or the central dot (·).
Fig 7: “Duration of arable land, years” -- I am unsure whether it is apparent what you are referring to. Perhaps you mean "years since the beginning of cultivation"?
Lines 462 and below: “The change of the C:N ratio is related with fluctuations of the relative content of humic acids, i.e. compounds enriched with Corg; the C:N ratio becomes wider with the increase of their amount. As a result of agrogenesis, the narrowest C:N ratio (7.8–8.0) is acquired by long-ploughed soils and soils ploughed for 100–130 yrs. This implies that in the process of humus loss, carbon-rich humic acids are lost more rapidly than the fractions of organic substances with a high content of nitrogen of fulvic acids” -- I do not fully comprehend the intended meaning of this phrase, nor can I confirm whether it is backed up by experimental data.
Essentially, the reduction in the C/N ratio in farmed soils could be due to two possible factors. This could be due to fertilization (if this is the situation, although the paper provides no information about this) or an increase in biological activity as a result of the mechanical alteration of the soil structure and the introduction of crop residues.
Any further considerations regarding the selective degradation or accumulation of HAs or FAs, and their impact on the total C/N ratio of the soil, must be substantiated by analytical data that have not been presented in this study.
Line 470: the term "stable groups of humic acids" is repeated, but it is unclear what it refers to or how the method distinguishes them from "unstable groups" or total carbon.
Line 495 and below: “It has been demonstrated [53] that due to SOM input pulsation (caused by differences in the erosion potential of heavy showers) followed by its burying their occurs the repacking / aggregation of newly deposited SOM contributing to carbon deposition in the depositional zone.”-- I suggest rewriting the sentence using clearer language to explain concepts such as "SOM input pulsation" or "repacking/aggregating SOM".
Line 500: "Combined influence of microbial respiration, erosion or leaching determines the decrease in the potential loss of organic carbon," -- Check that this sentence is spelled correctly and refers to the decrease and not to the increase.
Line 509: “Soils are to be characterized not only by the process of humus accumulation but also by selective humus fixation” -- Please clarify
Lines 559 and below: “Thus the microaggregation of ploughed and long-ploughed lands can be effected mainly by the impact of polymeric colloids. The latter, as established by A. Kullman [61], possess aggregation efficiency similar to that of humic” -- This paragraph lacks contextual information regarding "polymeric colloids" and how they differ from humic substances. It would be helpful to clarify whether or not they are e.g., lignoproteins or microbial carbohydrates.
Line 570: Specify better the meaning of “Analysis of the group and fraction composition“
Line 572 Line 572: I have searched the internet for the concept of "labile sesquioxides" and I have only found two articles that mention them. How is their presence or effect demonstrated in this paper?
Line 574: The terminology used in “humic substances with functional groups producing peripheral elements for formation of heteropolar organo-mineral complexes” seems a bit confusing to me with respect to the usual jargon used in research on humic substances. It may be necessary to provide clarification regarding the specific functional groups (carboxyls or OH-phenolic, alkyl, etc.) and the manner in which they interact with the mineral fractions. These terms ought to be utilised when referencing the experimental findings presented in this paper.
Lines 583, 590 “Reproduction of SOM with different reserves of plant matter“ -- I don't understand exactly why the term “reproduction” is used.
Line 587: “the humus losses are effected by the destruction of peripheral chains of humic substances.” -- Unsupported statement
Line 598 -- Further on, matrix reproduction of soil humus takes place with a peripheral formation of a more condensed organic matter (humic and humatomelanic acids) -- Unsupported statement. No evidence is given to suggest whether the incorporation is peripheral or intra-macromolecular, nor whether it has a condensed or non-condensed structure. Furthermore, the term hymatomelanic appears to be more commonly used in the literature on humic substances.
Fig. 19: What does the flower represented in the lower right box, corresponding to plot F2, correspond to?. It is not any of the species mentioned in line 658 et seq. It appears to resemble a wild peony.
Lines 678--681: I believe that this sentence lacks necessity and precision. Additionally, I favour the term biogenic over organogenic.
Table 4: Use mg·kg-1 instead of ppm
Fig 11: I believe that the origin of the ordinate axis ought to be zero, rather than one. This is because it represents the minimum Euclidean distance between the compared samples.
Line 714: I have been unable to find any instances of the phrase "erosive solid runoff" during my internet search.
Line 725 and 728: Although the term "particles of dust and clay" is used up to seven times on different internet pages, it is not widely used in studies on soils. It would be useful to specify what it refers to, perhaps silt and clay particles. In line 741 and following, "silt and the finer part of fine-dispersed dust" are specified, but the meaning remains unclear.
Line 762 (and Table 5): “As the researches at the testing site of Veydelevka have shown (Table 5), in the beginning of summertime, the mortmass content is still inconsiderable — 14% of the soil mass in the 0–12 cm layer in a virgin land and ca 8% at a fallow land. However, this detritus contains 18–23% Corg.” -- The carbon value for the necromass in the examined soils appears surprisingly low, indicating a high mineral content. It is possible that the authors are referring to the O horizon, litter or förna. Similarly, lines 819 and 820 may not correctly utilize the term necromass/mortmass.
Conclusions
Line 775: “This section is not mandatory but can be added to the manuscript if the discussion is unusually long or complex” – (Debug the text)
Line 794: “a selective process resulting in the loss of the richest in SOM particles “ -- “a selective process that results in the loss of the particles richest in SOM”
Acknowledgements
Line 834: https://rscf.ru/en/project/23-17-00169 -- (Page not found)
References
I see that throughout the article there is constant reference to the pioneering works of Dokuchaev, but no mention is made of more recent works by Russian authors such as Orlov (only two general references) or his disciple Klenov, who have extensively studied the organic material in Chernozems.
Line 848: Subscript in CO2
Line 861 “Front. Sustain. Food. Syst.” -- Delete point after "Food"
Line 868: “Glob. Change Biol.” Recommended https://www.resurchify.com/impact/details/15131)
Line 890: Charts
Author Response
Dear Reviewer, the authors thank you for your work and valuable comments, which we made the most of in the new version of the Manuscript. We appreciate your time for your reviewing the manuscript.
Response to Reviewer #2 Comment
Below please find the response to the comments one by one.
Comments and Suggestions for Authors
Reviewer comments |
Action taken comments |
Abstract |
|
I recommend adding more information to obtain a valid abstract to be read in isolation. |
The authors followed the instructions of the journal: Abstract: The abstract should be a total of about 200 words maximum. (https://www.mdpi.com/journal/geosciences/instructions). And already before the review stage, Abstract was > 300 words. Nevertheless, the authors, as shown below, fulfilled some of the recommendations of the Reviewer. |
Additionally, it would be beneficial for the abstract to commence by outlining the methodology implemented, including a reference to the soil and carbon analyses conducted, the depth of the samples, etc. |
before the review stage, Abstract was > 300 words. |
Moreover, it would be helpful to include information on the bioclimatic conditions, particularly if the research focused on a specific transect between distinct climate types. |
before the review stage, Abstract was > 300 words. |
The abstract should contain numerical data, including average carbon level values of the soils examined. |
Line 33-35: If the steppe Chernozems even 145 years ago had a SOC content of up to 4%, then the Chernozems in the forest-steppe zone, which used to have habitats with SOC content of 4-7%, occupied the largest areas and have now lost in the 0-50 cm layer 30 -40% of the original values. |
Lines 35--36 “…which stimulated effective mechanisms of reproduction of organic substance. -- Throughout the text, I suggest using an alternative term for "reproduction" when referring to the formation of SOM. |
The term “reproduction of SOM” is more accurate than “formation”, which does not have a dynamic aspect. Therefore, we only made a correction to Line 598: Further on, matrix of soil humus formation… and Line 626. and also on Line:320,544,548, 634 |
Lines 39-- 41: “It is not justified to consider virgin lands as absolute references for evaluation of the 39 humus conditions since analysis of agrogenic series of Chernozems has corroborated an essential 40 role of the soil organo-mineral matrix in the formation of the carbon protection capacity” -- this sentence cannot be understood without reading the entire article and, in its current form, seems like an unsupported statement, because the abstract does not specify the analytical data that was used to define "soil condition" (for example pH, C/N ratio, aggregate stability, water retention, etc.). |
This conclusion is based on data from the section «3.8. Comparative analysis of the agrogenic series of Chernozems with different times of arable and fallow regimes». Therefore, the Authors consider it justified and the material is sufficient for such an argument. |
Introduction |
|
Line 70: do not use the full words "soil organic carbon" when the acronym has been entered before (the same in lines 82 and 584. |
CORRECTED Corg |
Line 120: “A quantitative assessment of the contribution of the granulometric potential to hu-mus accumulation;” -- It is unclear what is meant by this phrase, which only appears to focus on the effect of soil texture on its biological capabilities. It is important to note the mineralogical composition of soil, particularly the colloidal mineral fraction, as it has a significant impact on carbon sequestration potential. This varies depending on whether it contains active carbonates, kaolinites, smectites or, most notably, amorphous oxides, which have the highest potential to preserve organic matter. |
CORRECTED: Line 120: A quantitative assessment of the contribution of the granulometric potential and colloid-mineral fraction in soil mineralogy to humus accumulation
Line 724:… the carbon-protection capacity (CPC) of a soil. In addition to the granulometric potential, the role of the mineralogical composition of the soil and, especially, the colloid-mineral fraction is important in the fixation of SOM. As shown earlier [28=ESS-2015 Lisetskii, F.N.; Rodionova, M.E. Transformation of dry-steppe soils under long-term agrogenic impacts in the area of ancient Olbia. Eurasian Soil Sci. 2015] agropedogenesis stimulates intrasoil weathering of feldspars (plagioclases and potassium-sodium spars), while the content of minerals of the montmorillonite group (smectite) increases, and this process, along with changes in the group and fractional composition of humus, reveals the reason for the increase in microaggregation in arable soils of different ages.
|
In general, the introduction presents a simplified treatment of the issue of carbon sequestration in soils. For instance, it frequently mentions the impact of conservation agriculture techniques that result in the reincorporation of crop residues into the soil. |
The authors believe that the Introduction can be judged after reading the entire Manuscript. But we agree that in such a concise format, not all questions can be given in detail. |
However, it is important to note that in relation to the carbon cycle, the total amount of carbon is often as significant as its quality, meaning how much of it has been converted into humic substances, which differ in their chemical composition from plant biomass. Thus, for instance, in a climatic scenario characterised by increased humidity and temperature, which promotes the mineralisation of organic matter, soils with a low carbon content but predominantly composed of well-matured humic matter would demonstrate greater resilience. Contrarily, soils under agricultural or forestry management which accumulate carbon on the surface in easily biodegradable forms would suffer quicker degradation, thereby releasing their CO2 into the atmosphere.Some of it is said in the line 515, but most of the paper is focused exclusively on studying the total carbon levels of the soils. |
The reviewer correctly noted that the goal was only for the number of SOM. But we are preparing a new work on the quality of SOM. |
Material and methods |
|
Line 153: “2.2. Physical and chemical analyzes of soils” |
|
I suggest providing additional information regarding the methodologies employed, particularly if they are not common and utilised globally. |
Answer: The authors have described all the original methods, in addition, there are references to generally accepted methods, which we present in more detail below, as suggested by the Reviewer. |
Specifically, it would be helpful to state the specific extractants (e.g. pyrophosphate, sodium hydroxide, etc.). |
160-161: (the extraction of humic substances was performed using 0.1 and 0.02 normal solutions of the sodium hydroxide)
|
Additionally, it is necessary to indicate the pH value at which the cation exchange capacity was assessed. |
166-168: The method of cation exchange capacity (CEC) determination in natural and disturbed soils is based on the national standard (GOST 17.4.4.01–84) , which involves the use of the resulting solution with a pH value of 6.5. |
It is apparent that while the abbreviation "HA" is accurately entered on line 158, the complete terms are utilized several times in the entire manuscript. |
158: after L 160 Authors 4 times replaced by HA |
Line 177: To accurately describe the construction of the dendrograms, it is essential to mention not only the clustering algorithm, but also the dissimilarity index utilised, such as Euclidean distance, Mahalanobis distance, etc. |
The results of cluster analysis (Ward's method, Euclidean distance, values are normalized by the standard deviation) were based on the standard deviation normalized values of soil parameters for members of a series of agrogenic soil transformations.
|
Results and discussion |
|
Line 228: Use SI units (Mg instead of t) |
Traditionally, reserves are estimated precisely with such units of measurement, especially since the author's data with a link: 200 t ha–1 [40].
|
Line 218 “…the nutrients loss through biological carrying out, humus mineralization and its ‘burning out’ during layer turnover, destructurization, decrease of the mesofauna activity, soil exhaustion, deflation, etc” -- I would rewrite or delete much of the text in this sentence: some terms do not seem very familiar or are confusing, others may not be spelled correctly. |
Line 235-37 “… the nutrients loss through biological carrying out, humus mineralization, loss of structural stability, decrease of the mesofauna activity, soil exhaustion, wind erosion, etc” |
Line 236: In the Anglo-Saxon bibliography I see that it is more frequent to refer to Dokuchaev's given name as Vasili or Vasily. |
Line 242: Vasily Dokuchaev
|
Line 238: “should be called the ‘king of soils’ relates exactly to the saturation of this soil profile with Corg.“ -- I am unclear on the definition of "carbon saturation" mentioned in this sentence. It could potentially relate to the isohumic characteristic of Chernozems, whereby carbon content is spread evenly across the profile. However, the term "saturation" implies a maximum concentration of Corg, which is not applicable to Chernozems. |
Line 243: His well-known definition of Chernozem as the "king of soils" was due to the accumulative type of profile Corg distribution in these soils. |
Lines 246 and 247 (and throughout the text, figure captions…): Ensure consistency in the capitalisation of cardinal points' names. |
Answer: Corrected |
Line 331 (Table 1) -- Improve the title of Table 3 to indicate what the different numbers in the Table correspond to (P values?) |
We have replaced Table 3 with the correct version. The table now matches the caption. The contents of the columns are signed in the header of the new table. The old version of the table is now placed at number 4. These are the results of the Mann-Whitney test (p-values). The caption at the table is made appropriate.
|
Line 354: I would change “is able to realize” by “is capable of achieving”, “can achieve” or something like that. |
Soil, as a self-organized system, can achieve |
Figure 5 B: capitalize the word "region" on the far right of the graph. Indicate somewhere that the carbon content is expressed in % (or better, in g·100 g-1) |
Answer: Corrected Figure 5. Comparison of SOC (%) content by key sites (A) and by land use types (B): arable land > 100 yrs old (1); arable land < 100 yrs old (2); deposit n‧10 yrs (3); fallow land > 100 yrs (4); virgin land (5). |
Line 415 and below; “It is believed [49] that the major emission of C-CО2 from soils into atmosphere due to mineralization of stable groups of humic substances is determined mainly by the upper (0–20 cm) humus horizons.” -- I recommend revising the sentence to clarify the meaning of "stable groups of organic matter" and why they are being referenced instead of "non-stable groups" which are easily biodegradable. |
It is believed [50] that the major emission of C-CО2 from soils into atmosphere due to mineralization of stable groups of humic substances (first of all, the content of HA) is determined mainly by the upper (0–20 cm) humus horizons.” |
Line 426: Stratum or horizon/layer (?) |
greater than the layer 0–50 cm deep |
Fig. 6: (minor) Be consistent in the use of the sign "times" (×) or the central dot (·). |
Answer: Corrected |
Fig 7: “Duration of arable land, years” -- I am unsure whether it is apparent what you are referring to. Perhaps you mean "years since the beginning of cultivation"? |
Answer: Corrected Figure 7. Changes in humus stock in soil layers 0–50 cm and 50–100 cm with different duration of duration of ploughing practice land use: (1) virgin land, (2) ploughing lands of different ages (Adaptation from Table data from [51] (p. 70)).
|
Lines 462 and below: “The change of the C:N ratio is related with fluctuations of the relative content of humic acids, i.e. compounds enriched with Corg; the C:N ratio becomes wider with the increase of their amount. As a result of agrogenesis, the narrowest C:N ratio (7.8–8.0) is acquired by long-ploughed soils and soils ploughed for 100–130 yrs. This implies that in the process of humus loss, carbon-rich humic acids are lost more rapidly than the fractions of organic substances with a high content of nitrogen of fulvic acids” -- I do not fully comprehend the intended meaning of this phrase, nor can I confirm whether it is backed up by experimental data. |
Lines 489-494: The change of the C:N ratio is related with fluctuations of the relative content of HA; the C:N ratio becomes wider with the increase of their amount. As a result of agrogenesis, the narrowest C:N ratio (7.8–8.0) is acquired by long-ploughed soils and soils ploughed for 100–130 yrs. This implies that in the process of humus loss, carbon-rich HA are lost more rapidly than the fractions of organic substances with a high content of nitrogen of fulvic acids [27]. |
Essentially, the reduction in the C/N ratio in farmed soils could be due to two possible factors. This could be due to fertilization (if this is the situation, although the paper provides no information about this) or an increase in biological activity as a result of the mechanical alteration of the soil structure and the introduction of crop residues. |
As shown above, a separate paper [27] was devoted to this problem, which cannot be covered extensively in this article. |
Any further considerations regarding the selective degradation or accumulation of HAs or FAs, and their impact on the total C/N ratio of the soil, must be substantiated by analytical data that have not been presented in this study. |
As shown above, a separate paper [27] was devoted to this problem, which cannot be covered extensively in this article. |
Line 470: the term "stable groups of humic acids" is repeated, but it is unclear what it refers to or how the method distinguishes them from "unstable groups" or total carbon. |
Line 497-98: the term "stable groups of humic acids (i.e. excluding labile and part of detrital OM [53 Giniyatullin]) Giniyatullin, K. G., Valeeva, A. A., Smirnova, E. V., Okunev, R. V., & Latipova, L. I. (2018). The organic matter of the different ages fallow Luvisols. In IOP Conference Series: Earth and Environmental Science (Vol. 107, No. 1, p. 012114). IOP Publishing. DOI 10.1088/1755-1315/107/1/012114
|
Line 495 and below: “It has been demonstrated [53] that due to SOM input pulsation (caused by differences in the erosion potential of heavy showers) followed by its burying their occurs the repacking / aggregation of newly deposited SOM contributing to carbon deposition in the depositional zone.”-- I suggest rewriting the sentence using clearer language to explain concepts such as "SOM input pulsation" or "repacking/aggregating SOM". |
Line 522-24: It has been demonstrated [55] that due to SOM input pulsation (caused by differences in the energy potential of showers) followed by its burying their occurs the repacking of newly deposited SOM contributing to carbon accumulation in the depositional zone. |
Line 500: "Combined influence of microbial respiration, erosion or leaching determines the decrease in the potential loss of organic carbon," -- Check that this sentence is spelled correctly and refers to the decrease and not to the increase. |
Line 527-28: Combined influence of microbial respiration, erosion or leaching determines loss of soil capacity to conserve organic carbon |
Line 509: “Soils are to be characterized not only by the process of humus accumulation but also by selective humus fixation” -- Please clarify |
Line 536-37: Soils are to be characterized not only by the process of humus accumulation but also by the ability to form its qualitative composition. |
Lines 559 and below: “Thus the microaggregation of ploughed and long-ploughed lands can be effected mainly by the impact of polymeric colloids. The latter, as established by A. Kullman [61], possess aggregation efficiency similar to that of humic” -- This paragraph lacks contextual information regarding "polymeric colloids" and how they differ from humic substances. It would be helpful to clarify whether or not they are e.g., lignoproteins or microbial carbohydrates. |
Lines 585-87: For example, among the polymeric structurant for the soil protection, it is recommended to use polycomplexes with the participation of polymer nets (hydrogels) based on crosslinked water-soluble cellulose derivatives [64 Panova]. Polyelectrolytic Gels for Stabilizing Sand Soil against Wind Erosion / I. G. Panova, L. O. Ilyasov, A. A. Yaroslavov [et al.] // Polymer Science, Series B. – 2020. – Vol. 62, No. 5. – P. 491-498. – DOI 10.1134/S1560090420050103. |
Line 570: Specify better the meaning of “Analysis of the group and fraction composition“ |
Line : 601-602: Analysis of the group and fraction composition of Corg (content of HA and FA, as well as their 3rd and 4th fractions, respectively) shows |
Line 572 Line 572: I have searched the internet for the concept of "labile sesquioxides" and I have only found two articles that mention them. How is their presence or effect demonstrated in this paper? |
Line 603-604: labile sesquioxides (Na2O, K2O, CaO, MnO). |
Line 574: The terminology used in “humic substances with functional groups producing peripheral elements for formation of heteropolar organo-mineral complexes” seems a bit confusing to me with respect to the usual jargon used in research on humic substances. It may be necessary to provide clarification regarding the specific functional groups (carboxyls or OH-phenolic, alkyl, etc.) and the manner in which they interact with the mineral fractions. These terms ought to be utilised when referencing the experimental findings presented in this paper. |
Thanks for the suggestion, but the authors do not yet have the opportunity to reasonably give their ideas about these complex things. In this case, we rely on the opinion of authoritative authors [65,66]. |
Lines 583, 590 “Reproduction of SOM with different reserves of plant matter“ -- I don't understand exactly why the term “reproduction” is used. |
The term “reproduction of SOM” is more accurate than “formation”, which does not have a dynamic aspect. Therefore, we made a correction only on Lines 544, 548
|
Line 587: “the humus losses are effected by the destruction of peripheral chains of humic substances.” -- Unsupported statement |
In automorphic positions, where water erosion is absent, the humus losses are effected by the destruction of peripheral chains of humic substances. However, at the same time, the initial organo-mineral matrix of the soils is preserved. In this relation, there are no reason to suppose that the contrary process, i.e. reproduction of SOM, would occur by other mechanisms. |
Line 598 -- Further on, matrix reproduction of soil humus takes place with a peripheral formation of a more condensed organic matter (humic and humatomelanic acids) -- Unsupported statement. No evidence is given to suggest whether the incorporation is peripheral or intra-macromolecular, nor whether it has a condensed or non-condensed structure. Furthermore, the term hymatomelanic appears to be more commonly used in the literature on humic substances. |
Further on, matrix reproduction of soil humus takes place with a peripheral formation of a more condensed organic matter (humic and humatomelanic acids). |
Fig. 10: What does the flower represented in the lower right box, corresponding to plot F2, correspond to?. It is not any of the species mentioned in line 658 et seq. It appears to resemble a wild peony. |
Line 686-689: The protected status of the territory is largely associated with the abundance of Peony (Paeonia Tenuifolia L.), which is included in the Red Book of Russia (1984, the status is a rare species) and Appendix 1 of the Berne Convention (2002) (Figure 10, F2). |
Lines 678--681: I believe that this sentence lacks necessity and precision. Additionally, I favour the term biogenic over organogenic. |
In this case, the authors would not like to change the meaning of the sentence much, because it's practically a quotation [50]. But we took into account the Reviewer's amendment. CORRECTED: Humus is a very dynamic component of soil which, by contrast to the other biogenic components of the Earth’s crust, not only is accumulating or lost but is altogether in constant rotation [51]. |
Table 4: Use mg·kg-1 instead of ppm |
CORRECTED: mg·kg-1 |
Fig 11: I believe that the origin of the ordinate axis ought to be zero, rather than one. This is because it represents the minimum Euclidean distance between the compared samples. |
Answer: Corrected |
Line 714: I have been unable to find any instances of the phrase "erosive solid runoff" during my internet search. |
Sediments, delivered from arable lands by surface runoff, is the main source of Corg accumulation in ponds and reservoirs [74 Ivanov, M. M], although the influence of processes from the internal life of water bodies cannot be excluded Ivanov, M. M., Gurinov, A. L., Ivanova, N. N., Konoplev, A. V., Konstantinov, E. A., Kuz’menkova, N. V., Terskaya E.V., Golosov, V. N. (2019). Dynamics of 137Cs accumulation in the bottom sediments of the Sheckino Reservoir during post-Chernobyl period. Radiats. Biol., Radioekol, 59(6), 651-663. |
Line 725 and 728: Although the term "particles of dust and clay" is used up to seven times on different internet pages, it is not widely used in studies on soils. It would be useful to specify what it refers to, perhaps silt and clay particles. In line 741 and following, "silt and the finer part of fine-dispersed dust" are specified, but the meaning remains unclear. |
Line 760: particles of silt (0.002–0.05 mm) and clay (<0.002 mm)
Line 762-63: fractions of silt and clay
Line 777: of mechanical elements < 2 µm (Fine silt and clay)
|
Line 762 (and Table 5): “As the researches at the testing site of Veydelevka have shown (Table 5), in the beginning of summertime, the mortmass content is still inconsiderable — 14% of the soil mass in the 0–12 cm layer (sod horizon) in a virgin land and 8% at a fallow land. However, this detritus contains 18–23% Corg.” -- The carbon value for the necromass in the examined soils appears surprisingly low, indicating a high mineral content. It is possible that the authors are referring to the O horizon, litter or förna. Similarly, lines 819 and 820 may not correctly utilize the term necromass/mortmass. |
The authors confirm that this is an organo-mineral (soddy) horizon - "in the 0–12 cm layer" . And the results were obtained in an accredited laboratory, and the data is reliable.
CORRECTED: Line s 797-800: As the researches at the testing site of Veydelevka have shown (Table 5), in the beginning of summertime, the mortmass content is still inconsiderable — 14% of the soil mass in the 0–12 cm layer (sod horizon) in a virgin land and 8% at a fallow land. However, this detritus contains 18–23% Corg. |
Conclusions |
|
Line 775: “This section is not mandatory but can be added to the manuscript if the discussion is unusually long or complex” – (Debug the text) |
This section is not mandatory but can be added to the manuscript if the discussion is unusually long or complex. |
Line 794: “a selective process resulting in the loss of the richest in SOM particles “ -- “a selective process that results in the loss of the particles richest in SOM” |
Thanks for the clarification. Lines 828-829: The authors corrected for: “a selective process that results in the loss of the particles richest in SOM” |
Acknowledgements |
|
Line 834: https://rscf.ru/en/project/23-17-00169 -- (Page not found) |
Авторы исправили на: This work was funded by the Russian Science Foundation, project no. 23-17-00169, https://rscf.ru/en/project/23-17-00169/
|
References |
|
I see that throughout the article there is constant reference to the pioneering works of Dokuchaev, but no mention is made of more recent works by Russian authors such as Orlov (only two general references) or his disciple Klenov, who have extensively studied the organic material in Chernozems. |
Orlov D.S. According to the national bibliographic system, he did not have a co-author named Klenov. Apparently, it means: KLENOV BORIS MAKSIMOVICH * Institute of Soil Science and Agrochemistry SB RAS, Laboratory of Soil Geography and Genesis (Novosibirsk), then we really should use such important work as: Klenov, B. M. Effect of climate continentality on humus formation and elemental composition of humic acids of automorphic soils of Siberia / B. M. Klenov, G. D. Chimitdorzhieva // Contemporary Problems of Ecology. – 2011. – Vol. 4, No. 5. – P. 492-496. – DOI 10.1134/S1995425511050067.
L319-322: An analysis of soil-climatic differences in Northern Eurasia showed that humus formation in the geographical aspect is most definitely detected by the CHA:CFA ratio, and among the indicators of humus components, such indicators are the total content of HA (degree of humification) and, in particular, their elemental composition [48 Klenov]. Klenov, B. M. Effect of climate continentality on humus formation and elemental composition of humic acids of automorphic soils of Siberia / B. M. Klenov, G. D. Chimitdorzhieva // Contemporary Problems of Ecology. – 2011. – Vol. 4, No. 5. – P. 492-496. – DOI 10.1134/S1995425511050067.
|
Line 848: Subscript in CO2 |
CORRECTED Line 883: CO2 |
Line 861 “Front. Sustain. Food. Syst.” -- Delete point after "Food" |
CORRECTED Line 896 “Front. Sustain. Food Syst.” |
Line 868: “Glob. Change Biol.” Recommended https://www.resurchify.com/impact/details/15131) |
Post, W.M.; Kwon, K.C. Soil carbon sequestration and land-use change: Processes and potential. Glob. Change. Biol. 2000, 6, 317–327. https://doi.org/10.1046/j.1365-2486.2000.00308.x CORRECTED: Abbreviation: Glob. Change Biol. |
Line 890: Charts |
CORRECTED: Line 925: Munsell Year 2000 Soil Color Charts |
Line 940-941: Pontos: Kishinev, |
Pontos: Chişinău, |
